



# Seasonal overturning variability in the eastern North Atlantic subpolar gyre: A Lagrangian perspective

Oliver J. Tooth[1], Helen L. Johnson[1], Chris Wilson[2], and Dafydd G. Evans[3]

[1]Department of Earth Sciences, University of Oxford, Oxford, United Kingdom
[2]National Oceanography Centre, Liverpool, United Kingdom
[3]National Oceanography Centre, Southampton, United Kingdom

**Correspondence:** Oliver J. Tooth (oliver.tooth@seh.ox.ac.uk)

**Abstract.** Changes in the high-latitude Atlantic Meridional Overturning Circulation (MOC) are dominated by water mass transformation in the eastern North Atlantic Subpolar Gyre (SPG). Both observations and ocean reanalyses show a pronounced seasonality of the MOC within this region. Here, we investigate the nature of this seasonal overturning variability within the eastern SPG using Lagrangian water parcel trajectories evaluated within an eddy-permitting ocean sea-ice hindcast simulation. Our analysis highlights the critical role of water parcel recirculation times in determining the seasonality of overturning measured in both the traditional Eulerian and complimentary Lagrangian frames of reference. From an Eulerian perspective, we show that the minimum of the MOC seasonal cycle in autumn results from a combination of enhanced stratification and increased southward transport within the upper East Greenland Current. This convergence of southward transport within the MOC upper limb is explained by decreasing water parcel recirculation times in the upper Irminger Sea, consistent with a gyre-scale response to seasonal wind forcing. From a Lagrangian perspective, we find that upper limb water parcels flowing northwards into the eastern SPG participate in a recirculation race against time to avoid wintertime diapycnal transformation into the lower limb of the MOC. The majority of water parcels, sourced from the central and southern branches of the North Atlantic Current, are unsuccessful and thus determine the mean strength of overturning within the eastern SPG (8.9 ± 2.2 Sv). The seasonality of Lagrangian overturning is explained by a small collection of upper limb water parcels, recirculating rapidly (≤ 8.5 months) in the upper Irminger and Central Iceland Basins, whose along-stream transformation is dependent on their time of arrival in the eastern SPG.

## 1 Introduction

The Atlantic Meridional Overturning Circulation (MOC) plays a critical role in the global climate system through the uptake and redistribution of heat, freshwater and anthropogenic carbon (McKinley et al., 2017; Bryden et al., 2020; Fay and McKinley, 2021; Li et al., 2021b). In the Subpolar North Atlantic (SPNA), the MOC is characterised by the transformation of warm, saline subtropical waters, transported northward in the North Atlantic Current (NAC), into cold, dense, carbon-rich waters flowing southward at depth (Buckley and Marshall, 2016). Traditionally, the formation of dense water masses feeding the lower limb of the MOC has been ascribed to open-ocean deep convection in the Labrador and Nordic Seas (e.g., Biastoch et al. 2008; Medhaug et al. 2012; Menary et al. 2015; Heuzé 2017; Li et al. 2019). However, a combination of both theoretical and





observational studies have challenged this long-standing paradigm (Spall, 2004; Pickart and Spall, 2007; Wåhlin and Johnson, 2009; Holte and Straneo, 2017), prompting a re-examination of the locations and mechanisms governing variations in the strength of subpolar overturning.

The Overturning in the Subpolar North Atlantic Program (OSNAP) was initiated in summer 2014 to complement the Rapid Climate Change - Meridional Overturning Circulation Heatflux Array (RAPID-MOCHA) along 26.5°N (Cunningham et al., 2007) by providing the first continuous measurements of the MOC at subpolar latitudes (Lozier et al., 2017). The OSNAP observing system comprises two trans-basin sections which extend across the Labrador Sea (OSNAP West) and the Irminger and Iceland Basins (OSNAP East) (Lozier et al., 2017; Li et al., 2017). The strength of the MOC is diagnosed directly from the volume transports and hydrography measured along the OSNAP array, thereby translating water mass transformation along the complex circulation pathways of the Subpolar Gyre (SPG) and the Nordic Seas overflows into a single quantity, zonally integrated at the basin-scale (Lozier et al., 2019; Li et al., 2021a). In contrast with the subtropics (Frajka-Williams et al., 2019), the subpolar MOC is more appropriately calculated in density coordinates as the total transformation from lighter to denser water masses north of the OSNAP array (Lherminier et al., 2007; Holliday et al., 2018; Lozier et al., 2019). This is because a substantial portion of subpolar overturning is due to the horizontal circulation of the SPG across sloping isopycnal surfaces (Zhang and Thomas, 2021; Hirschi et al., 2020), which cannot imprint onto the MOC calculated in depth coordinates.

The OSNAP data reveal that the mean strength and variability of the MOC are dominated by dense water formation east of Greenland, rather than in the Labrador Sea. Further investigation of the sources of North Atlantic Deep Water (NADW) feeding the MOC lower limb have shown that dense water formation is distributed approximately equally between the Iceland and Irminger Basins and the Nordic Seas (Bringedal et al., 2018; Chafik and Rossby, 2019; Petit et al., 2020). The deepest portion of the MOC lower limb, known as lower NADW, is sourced from dense overflow waters spilling over the Greenland-Scotland Ridge (Dickson and Brown, 1994; Eldevik et al., 2009). Overflow waters are formed by a combination of continuous surface buoyancy loss along the perimeter of the Nordic Seas (Mauritzen, 1996; Isachsen et al., 2007; Huang et al., 2021) and wintertime open-ocean convection in the basin interior (Våge et al., 2011, 2015). Open-ocean convection also occurs in the southwestern Irminger Sea (Våge et al., 2008; Piron et al., 2016), resulting in the formation of a water mass with a similar composition to deep Labrador Sea Water (LSW) (Pickart et al., 1997; Rhein et al., 2015), known as deep Irminger Sea Intermediate Water (ISIW) (Le Bras et al., 2020). Upper ISIW, constituting the lightest component of the MOC lower limb (Le Bras et al., 2020), is instead formed by the progressive densification of Subpolar Mode Waters (SPMW) advected along the cyclonic pathways of the eastern SPG (Brambilla and Talley, 2008; Thierry et al., 2008). Crucially, it is variations in the production of upper and deep ISIW south of the Greenland-Scotland Ridge which dominates the variability of the subpolar MOC on monthly to decadal timescales (Petit et al., 2020; Desbruyères et al., 2019).

Throughout the North Atlantic Ocean, observational and modelling studies demonstrate that the variability of the MOC is larger on seasonal than interannual-decadal timescales (Willis, 2010; Mielke et al., 2013; Xu et al., 2014). At subpolar latitudes, observations reveal a pronounced seasonality in the MOC (Mercier et al., 2015; Li et al., 2021a), which is closely related to the formation and export of seasonal western boundary density anomalies (Holte and Straneo, 2017; Le Bras et al., 2020). In particular, the recent modelling study of Wang et al. (2021) attributed the overturning seasonality at OSNAP East to the



projection of seasonal density changes in the upper Irminger Sea onto the mean barotropic transport of the East Greenland
      Current (EGC). Since the formation of dense waters exported southward in the EGC is principally governed by surface heat
      loss over the Iceland and Irminger Basins (Brambilla and Talley, 2008; Petit et al., 2020; Pacini et al., 2020), we might
      anticipate a seasonal cycle of overturning which closely reflects the seasonality of surface buoyancy forcing north of OSNAP
      East. However, the observed amplitude of seasonal overturning variability ($\sim$4 Sv where 1 Sv $\equiv$ 1x10$^6$ m$^3$ s$^{-1}$; Mercier et al.
2015; Li et al. 2021a) is typically 5 times smaller than the magnitude of seasonal surface-forced transformation within the
      eastern SPG ($\sim$20 Sv; Xu et al. 2018a; Petit et al. 2020). While this difference is partially explained by large, compensating
      seasonal volume changes of the lightest water masses (Brambilla et al., 2008; Evans et al., 2017), overturning seasonality also
      remains small compared with the observed mean strength of the MOC ($\sim$16 Sv) in the eastern SPNA (Sarafanov et al., 2012;
      Mercier et al., 2015; Lozier et al., 2019; Li et al., 2021a; Chafik and Rossby, 2019).

In addition to seasonal variations in the formation of dense water north of OSNAP East, the recent study of Le Bras et al.
      (2020) demonstrates that seasonal overturning variability is closely related to the timescales of subduction and export within
      the western boundary current of the Irminger Sea. In particular, the authors showed that variations in the formation rate of
      upper ISIW formed in the vicinity of the boundary current can contribute significantly to overturning seasonality since upper
      ISIW is rapidly exported across OSNAP East in several months. Meanwhile, variability in the formation rate of deep ISIW
formed in the Irminger Sea interior projects onto interannual overturning variability owing to its slower entrainment into the
      western boundary current (Le Bras et al., 2020). The dominant role of upper ISIW in driving seasonal overturning variability at
      OSNAP East has since been further highlighted by Li et al. (2021a), who revealed strong seasonality in the observed thickness
      of upper ISIW in both the EGC and the Irminger Sea interior between 2014-2018. While the aforementioned studies have
      successfully identified the final water mass properties associated with seasonal overturning variability in the eastern SPNA, we
have yet to establish the origins of seasonally transformed water masses along the OSNAP East section. It therefore remains an
      open question whether there is a clear distinction between the pathways and advective timescales of water masses contributing
      substantially to the mean strength of the MOC compared with those responsible for its variability on seasonal timescales.

          In this study, we investigate the nature of seasonal overturning variability in the eastern SPG using a Lagrangian particle
      tracking tool in conjunction with an eddy-permitting, global ocean sea-ice hindcast simulation. Extending the Lagrangian
overturning framework recently introduced by Tooth et al. (2022), we use water parcel trajectories, initialised in the northward
      inflows across a model-defined OSNAP East array, to determine the distribution of seasonal overturning variability across the
      cyclonic circulation pathways of the eastern SPG. Moreover, we demonstrate the critical role of water parcel recirculation times
      within the eastern SPG in shaping the seasonal cycles of both the Lagrangian and Eulerian overturning evaluated at OSNAP
      East.

The manuscript is structured as follows. We begin by introducing the numerical model simulation, Lagrangian particle
      tracking approach, and defining the overturning in density-space at OSNAP East in Section 2. The simulated seasonal variability
      of the MOC at OSNAP East is explored from both Eulerian and Lagrangian perspectives in Section 3. Section 4 diagnoses the
      origins and advective timescales of water parcels contributing to the mean strength and seasonality of Lagrangian overturning.
      In Section 5, we decompose the seasonal cycle of Lagrangian overturning by circulation pathway, and examine the role of



along-stream diathermal and diahaline transformations in driving seasonal water mass modification. Section 6 addresses the mechanisms of seasonal Eulerian overturning variability, including the important role of water parcel recirculation times in the upper Irminger Sea. The manuscript concludes with a discussion of our principal findings in Section 7.

## 2  Model and methods

### 2.1  Model description

In this study we use output from the ORCA025-GJM189 ocean-sea-ice hindcast simulation produced by the Drakkar initiative (Barnier et al., 2006). The simulation uses a global implementation of the Nucleus for European Modeling of the Ocean (NEMO) model coupled to the thermodynamic Louvain-la-Neuve Ice Model version 2 (LIM2) (Fichefet and Morales Maqueda, 1999). The ocean is simulated using the eddy-permitting ORCA025 configuration of the NEMO v3.5 model (Madec, 2014), which utilises the ORCA tripolar grid configured with a nominal horizontal resolution of 1/4° (27.75 km at the equator, ∼12 km

in the Arctic) and 75 unevenly spaced z-coordinate levels in the vertical. Sub-grid scale parameterisations include horizontal biharmonic viscosity for momentum, Laplacian isopycnal diffusivity for tracers, and the TKE turbulent closure scheme for vertical mixing (Barnier et al., 2006). A comprehensive description of the ORCA025-GJM189 configuration can be found in Molines (2021).

The ORCA025-GJM189 hindcast used here simulates the historical period from 1958-2015, initialised from rest. Atmo-

spheric forcing is computed using the CORE bulk formulae and the Drakkar Forcing Set 5.2, which combines surface fluxes from the ERA40 (Uppala et al., 2005) and ERA-Interim reanalyses (Dee et al., 2011). The initial conditions of the simulation are provided by a combination of the Levitus climatological hydrography (Levitus et al., 1998) and the Polar Science Center Hydrographic Climatology (Steele et al., 2001) in the Arctic. A relaxation of sea surface salinity to the Levitus climatological hydrography with a piston velocity of 167 mm day$^{-1}$, equivalent to a 60 day decay time for 10 m of water depth

(Molines, 2021), is included to minimise model drift. Following the approach of MacGilchrist et al. (2020), the first 18 years are discarded, to allow for spin-up. The period 1976-2015 is analysed using 5-day model mean output.

### 2.2  Lagrangian particle tracking

To evaluate the trajectories of numerical water parcels advected by the time-varying velocity fields of the ORCA025-GJM189 hindcast, we use the Lagrangian offline particle tracking tool TRACMASS (v7.1, Aldama-Campino et al. 2020). TRACMASS

determines the trajectory of each water parcel using a stepwise-stationary scheme, which divides the time between successive 5-day mean velocity fields into a series of intermediate time steps. The velocity field at each intermediate time step is determined by linear interpolation and is assumed to be steady for the duration of the step (Döös et al., 2017). A water parcel's trajectory path through each model grid cell can therefore be determined analytically whilst conserving the mass contained within each grid cell (Döös et al., 2013). As such, and because the ocean model is Boussinesq, the volume transport conveyed by each



particle is conserved along its entire trajectory. For a comprehensive description of TRACMASS and its associated trajectory schemes readers are referred to Döös et al. (2017).

We make use of the Lagrangian experiment documented in Tooth et al. (2022), in which numerical water parcels are initialised in the northward inflows across the OSNAP East section (defined as in Menary et al. 2020) at the earliest available day of each month (based on the centre of the nearest 5-day mean window) for a period of 33 years (1976-2008). In total, more than
11.2 million water parcels are initialised, sampling the entire northward transport across OSNAP East over 396 months. During each monthly initialisation, the strategy for particle release is to distribute particles in proportion to the volume transport through each grid cell, with a minimum of one particle per cell. An increment of 2.5 mSv per particle, per cell is used. Once initialised, water parcels are advected solely by the time-evolving velocity field except when found inside the surface mixed layer. Here, we parameterise the effects of vertical convective mixing by introducing random vertical displacements along
water parcel trajectories (Paris et al., 2013), governed by a maximum vertical velocity of $|w| = 10$ cm s$^{-1}$ following Georgiou et al. (2021). Water parcels are removed from the experiment on meeting any one of three conditions: (i) on returning to the OSNAP East section (red trajectories in Fig. 1a), (ii) on crossing the Greenland-Scotland Ridge northwards (green trajectories in Fig. 1a), or (iii) upon reaching the maximum advection time of 7 years. Our decision to terminate water parcels on reaching the Greenland-Scotland Ridge is motivated by recent observations, which show that monthly overturning variability at OSNAP
East is dominated by surface buoyancy forcing over the Iceland and Irminger Basins rather than by variations in the overflow transport exiting the Nordic Seas (Bringedal et al., 2018; Østerhus et al., 2019; Petit et al., 2020). It should also be noted that > 99.1% of water parcels are intercepted on crossing either OSNAP East or the Greenland-Scotland Ridge within the 7-year maximum advection time. The remaining 0.9% of water parcels almost exclusively circulate at depth within the lower limb of the MOC north of OSNAP East and therefore do not contribute to the strength of overturning.

## 145 2.3 Definitions of the overturning in density-space

We use two complementary measures of the overturning in potential density-space at OSNAP East in this study. The Eulerian overturning streamfunction, $\psi(\sigma, t)$, is calculated directly from the time-evolving velocity and potential density fields along the OSNAP East section following Lozier et al. (2019):

$$\psi(\sigma, t) = \int\limits_{\sigma_{min}}^{\sigma} \int\limits_{x_w}^{x_e} v(x, \sigma, t) \, dx \, d\sigma \tag{1}$$

where $v(x, \sigma, t)$, the velocity normal to the model-defined OSNAP East section (positive values represent northward transports), is first integrated from the east coast of Greenland ($x_w$) to the Scottish shelf ($x_e$). We then obtain the Eulerian overturning streamfunction with respect to potential density, $\psi(\sigma, t)$, by integrating from the sea surface ($\sigma_{min}$) to a particular isopycnal layer ($\sigma$).

We denote the isopycnal at which the Eulerian overturning streamfunction reaches a maximum as $\sigma_{MOC}$, representing the
interface between the upper and lower limbs of the MOC at OSNAP East. The maximum overturning, measuring the overall



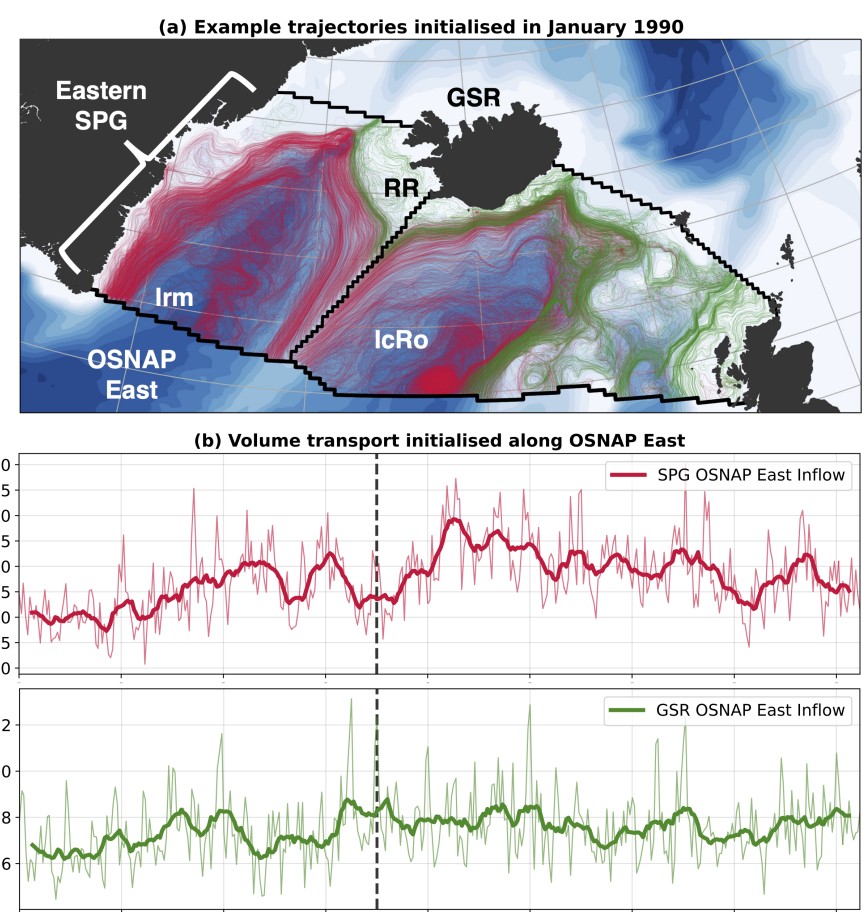

**Figure 1.** Lagrangian trajectories and volume transports across OSNAP East. (a) Example Lagrangian trajectories initialised on the northward inflows across OSNAP East in January 1990. Trajectories of water parcels which recirculate back to OSNAP East within the eastern SPG (south of the Greenland-Scotland Ridge) are shown in red, whereas green trajectories represent water parcels which cross the Greenland-Scotland Ridge northwards as Atlantic inflows to the Nordic Seas. (b) Monthly volume transports initialised on the northward inflows across OSNAP East which recirculate within the eastern SPG (red, upper panel) and those which cross the Greenland-Scotland Ridge northwards (green, lower panel). The bold lines overlaid are low pass filtered volume transports using a 1-year running mean. The grey dashed line indicates the volume transports across OSNAP East in January 1990.

strength of the MOC, therefore equates to the net northward transport within the upper limb integrated along the section. To ensure comparability between our analyses, Eulerian overturning streamfunctions are computed on the earliest available day of each month between 1976-2008.

To complement the Eulerian density-space overturning, we additionally diagnose the strength of the overturning at OSNAP East using a Lagrangian measure recently introduced by Tooth et al. (2022). The Lagrangian Overturning Function (LOF),



$F(\sigma, t, \tau_{max})$, quantifies the total light-to-dense transformation occurring along water parcel trajectories flowing northward across OSNAP East at the earliest available day of each month. To compute $F(\sigma, t, \tau_{max})$, we first extract the 9.8 million water parcels which recirculate back to OSNAP East within the eastern SPG (south of the Greenland-Scotland Ridge) to form 396 trajectory ensembles, one for each month between 1976-2008. For each monthly ensemble containing $N$ water parcels, the LOF is given by:

$$F(\sigma, t, \tau_{max}) = \int_{\sigma_{min}}^{\sigma} V_{North}(\sigma, t) - V_{South}(\sigma, t + \tau_{max}) \, d\sigma \qquad (2)$$

where $V_{North}$ represents the volume transport distribution of the ensemble water parcels in density-space on their northward crossing of the OSNAP East section at time $t$. This is calculated by integrating the absolute volume transports ($V_n$) of the $N$ water parcels in discrete potential density bins, where the bin width $\Delta\sigma$ is prescribed as 0.01 kg m$^{-3}$. Once the ensemble of water parcels have been advected for a maximum of $\tau_{max} = 7$ years, we construct an equivalent volume transport distribution in density-space, $V_{South}$, using the properties of water parcels on their southward crossing of the OSNAP East section at time $t + \tau$, where $0 < \tau \leq \tau_{max}$. Since the volume transport conveyed by each water parcel is conserved along its trajectory, the total northward and southward transports across OSNAP East due to recirculating water parcels are equal in the Lagrangian overturning calculation, and thus there is no net flow across the section. The LOF for each monthly ensemble is obtained by taking the cumulative sum of the net volume transport distribution (i.e., $V_{North}(\sigma, t)$ - $V_{South}(\sigma, t + \tau_{max})$) from the surface to the ocean floor. In keeping with its Eulerian counterpart, the strength of Lagrangian overturning (LMOC) is given by the maximum of the LOF, which occurs at the isopycnal of maximum Lagrangian overturning $\sigma_{LMOC}$.

## 3 Seasonal overturning variability at OSNAP East

We begin by exploring the seasonal variations in the density-space overturning at OSNAP East in ORCA025-GJM189 from both Eulerian and Lagrangian perspectives.

### 3.1 The Eulerian perspective

Over the duration of the study period (1976-2008), the time-mean strength of the monthly MOC at OSNAP East is 16.6 ± 2.7 Sv in ORCA025-GJM189, corresponding closely with observed estimates of the MOC at both 59.5°N (16.6 ± 1.1 Sv in 2002-2008; Sarafanov et al. 2012) and OSNAP East (16.8 ± 0.6 Sv in 2014-2018; Li et al. 2021a). The mean isopycnal of maximum Eulerian overturning ($\sigma_{MOC}$ = 27.52 kg m$^{-3}$) is lighter in the simulation compared with observations (27.55 kg m$^{-3}$), owing to the shoaling of the SPNA overturning cell in response to the excessive entrainment of ambient Atlantic water by the Nordic seas overflows in the model (MacGilchrist et al., 2020). As shown in Tooth et al. (2022), the simulated mean net throughflow to the Arctic Ocean at OSNAP East is 1.2 ± 1.1 Sv, in close agreement with the 1.0-1.6 Sv required by inverse models (Li et al., 2017; Lherminier et al., 2007). On subtracting this net throughflow from the mean strength of the MOC, we find that, on average, 15.4 Sv of upper limb waters are overturned north of OSNAP East in this simulation.



Figure 2a shows that the strength of the MOC at OSNAP East exhibits variability on monthly to interannual timescales. Concordant with the previous studies of Lozier et al. (2019) and Mercier et al. (2015), we find that MOC variability in the eastern SPNA is most pronounced on monthly timescales (SD = $\pm 2.7$ Sv), where monthly MOC values range from 7.8 Sv in October 1978 to 28.7 Sv in January 1996. The simulated MOC variability at OSNAP East is weaker on interannual timescales

(SD of annual means = $\pm 1.0$ Sv), in agreement with previous results from ocean models and reanalyses (Xu et al., 2014; Wang et al., 2021). The gradual increase in the MOC from the 1970s to the mid-1990s (+0.1 Sv yr$^{-1}$, $p < 0.01$) and the subsequent marked decline in the MOC between 1996 and 2000 (-1.3 Sv yr$^{-1}$, $p < 0.01$) are well-documented trends, consistently found in both observations (Kieke et al., 2007; Mercier et al., 2015) and numerical modelling studies (Böning et al., 2006; Desbruyères et al., 2013; Xu et al., 2013).

The intra-annual variability in the MOC at OSNAP East is dominated by the strong seasonal cycle shown in Figure 2b. The peak-to-peak amplitude of the seasonal cycle is 4.1 Sv, in close correspondence with the 4.2 Sv found by Wang et al. (2021) using a 1/12° Global Ocean Physics Reanalysis (1993-2016) and the 4.3 Sv seasonal cycle recorded along the OVIDE section (1993-2010) by Mercier et al. (2015). Figure 2b indicates that, on average, the MOC reaches a maximum in April (18.5 Sv) and a minimum in October (14.4 Sv). We note that, while there is strong consensus that the subpolar MOC reaches a maximum

in spring (e.g. Holte and Straneo 2017; Li et al. 2021a), the occurrence of the seasonal MOC minimum during autumn in this simulation disagrees with the findings of previous reanalysis (August; Wang et al. 2021) and observational (December; Mercier et al. 2015) studies. One possible reason for this inconsistency between studies is that the MOC seasonal cycle, when computed from monthly composites, is non-stationary, such that the timing of the seasonal extrema will be dependent upon the years chosen to compute those composites. This is particularly notable in the results of Wang et al. (2021), where the

most probable timing of the minimum of the MOC seasonal cycle at OSNAP East transitions from autumn during the years overlapping our study period (1993-2008) to summer between 2009-2016.

Given that the strength of the MOC equates to the net transport above $\sigma_{MOC}$, seasonal variations in the MOC can be attributed to either changes in the meridional velocity field or to changes in the isopcynal structure at OSNAP East. Figure 2b shows that $\sigma_{MOC}$ exhibits a pronounced seasonal cycle (peak-to-peak amplitude is 0.2 kg m$^{-3}$) at OSNAP East, which lags

seasonality in the strength of the MOC by 1 month. Although often overlooked, seasonal variations in $\sigma_{MOC}$ provide valuable insight into how the composition of the MOC upper limb evolves on intra-annual timescales. To explore this further, Figure 2c presents monthly composites of the Eulerian overturning streamfunction evaluated at OSNAP East. It should be noted that the maximum values of the monthly mean streamfunctions presented in Figure 2c underestimate the mean strength of the MOC determined from the maxima of monthly mean streamfunctions in Figure 2b (Lozier et al., 2019). This is because the

isopycnal of maximum overturning, $\sigma_{MOC}$, can vary substantially within each of the monthly composites. Figure 2c shows that the maximum of the MOC seasonal cycle in April occurs when upper limb waters flowing northwards across OSNAP East are relatively cold and dense ($\sigma_{MOC}$ = 27.57) having experienced intense surface buoyancy loss along the NAC during the previous winter (Grist et al., 2016). Conversely, the minimum of the MOC seasonal cycle in October is characterised by the arrival of warm and light upper limb waters ($\sigma_{MOC}$ = 27.50) which have been transformed by summertime buoyancy gain in

the NAC during the months prior (Li et al., 2021b).



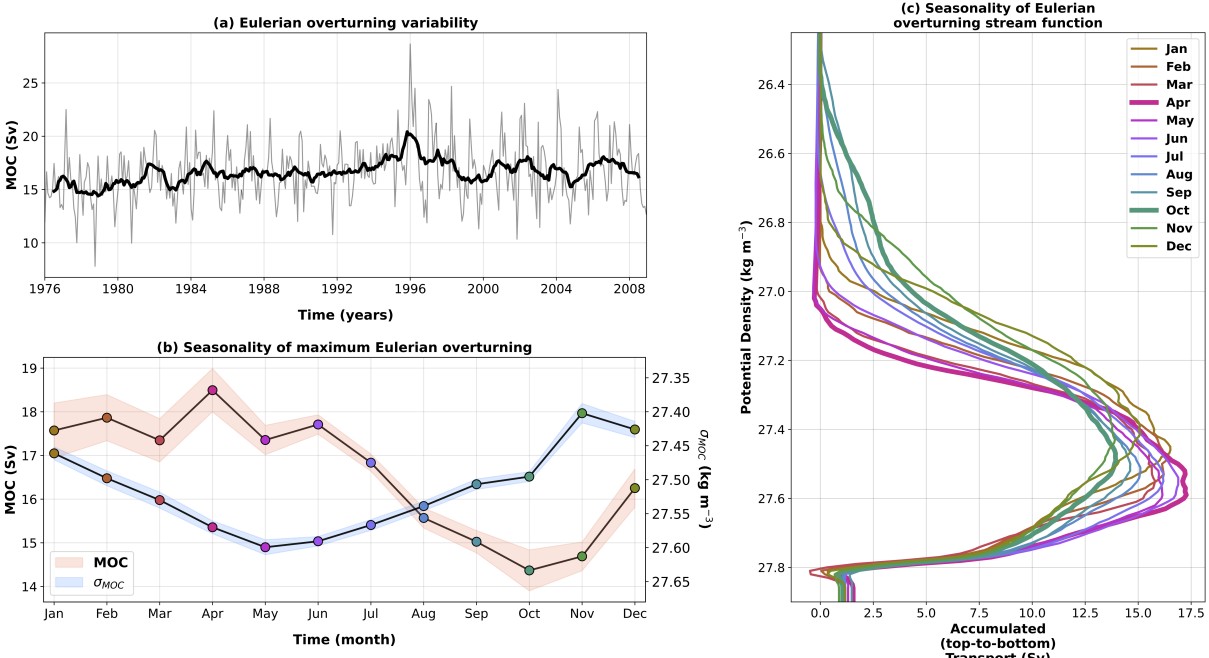

**Figure 2.** Eulerian overturning variability at OSNAP East. (a) Monthly maximum of the Eulerian overturning streamfunction in density-space (first available day of each month, grey line) and 1 year annual running mean (black bold line) overlaid for 1976-2008. (b) Mean seasonal cycles of the maximum of the Eulerian overturning streamfunction and the isopycnal of maximum overturning ($\sigma_{MOC}$) computed from monthly composites. Shading represents $\pm 1$ standard error of the monthly estimates. (c) Monthly composites of the Eulerian overturning streamfunction calculated in discrete density-space (bin width is 0.01 kg m$^{-3}$); maximum (April) and minimum (October) months in the MOC seasonal cycle are included in bold.

The seasonal cycle in the strength of the MOC at OSNAP East therefore corresponds to the expansion (MOC weakening in summer-autumn) and contraction (MOC strengthening in winter-spring) of the overturning streamfunction in density-space, which is consistent with seasonal variations in surface buoyancy forcing over the NAC upstream.

## 3.2 The Lagrangian perspective

The Lagrangian overturning framework provides us with an alternative view of the overturning variability at OSNAP East on seasonal timescales. Whereas the Eulerian streamfunction integrates the meridional transports across OSNAP East in density-space at a given point in time, the LOF measures the net diapycnal transformation that the total northward transport arriving at OSNAP East will go on to experience during its recirculation within the eastern SPG. As such, the meridional transports comprising the LOF belong to a single collection of water parcels (sharing a common inflow time), whereas the Eulerian 235 streamfunction includes two unrelated collections of water parcels flowing northwards and southwards, respectively. We should




therefore consider the LOF to be a complementary measure of the overturning at OSNAP East, which preserves knowledge of water parcel identity at the expense of integrating across water parcel recirculation times which can extend from days to years.

Figure 3a presents the strength of the LMOC within the eastern SPG between 1976 and 2008, consistent with our earlier Eulerian analysis (Fig. 2a). We find that, on average, 8.9 Sv of transport flowing northwards across OSNAP East is transformed
from the upper to the lower limb south of the Greenland-Scotland Ridge. Importantly, this transformation is stronger than the mean eastern SPG Lagrangian overturning found in Tooth et al. (2022), because here we compute the average of the maximum Lagrangian overturning each month as opposed to taking the maximum of the mean LOF as in Tooth et al. (2022). We attribute the remaining 6.5 Sv (15.4 Sv - 8.9 Sv) of overturning at OSNAP East to water parcels which are transformed north of the Greenland-Scotland Ridge before returning via the deep pathways of Nordic Seas overflows. Although we do
not resolve the overturning pathways of the Nordic Seas overflows in this Lagrangian experiment, the close correspondence between the month-to-month Lagrangian overturning variability of the eastern SPG (SD = $\pm$2.2 Sv) shown in Figure 3a and the total month-to-month variability of the MOC (SD = $\pm$2.7 Sv) underscores the dominant contribution made by the eastern SPG to intra-annual overturning variability at OSNAP East.

In contrast with the seasonality of the Eulerian MOC, the seasonal cycle of Lagrangian overturning within the eastern SPG
shows a steady increase from a minimum of 6.4 Sv in May to a maximum of 11.5 Sv in November. While the phase difference between the seasonal cycles of the MOC and the LMOC may initially appear counter-intuitive, recall that the strength of the Lagrangian overturning in Figure 3b quantifies how much of the total northward transport initialised along OSNAP East each month is transformed from the upper to the lower limb in the Irminger and Iceland-Rockall Basins. Interestingly, the seasonal cycle of the LMOC strength corresponds closely with the seasonality of both $\sigma_{LMOC}$ and $\sigma_{MOC}$ (see Figs. 2b and
3b), suggesting that the potential density of upper limb water parcels on flowing northward across OSNAP East is an important indicator of their future contribution to the overturning north of the section. This relationship can also be seen in Figure 3c, which shows how the LOF evolves over the duration of the mean seasonal cycle. Here, we find that the maximum of the LMOC in November occurs when the largest number of relatively light upper limb water parcels are able to integrate sufficient wintertime surface buoyancy loss to enter the lower limb before returning to OSNAP East. Meanwhile, in May, we find that
denser upper limb waters, previously transformed by wintertime cooling in the NAC, experience substantial summertime buoyancy gain north of OSNAP East to become lighter downstream. This negative diapycnal transformation manifests in Figure 3c through both the pronounced region of negative Lagrangian overturning between 26.8-27.2 kg m$^{-3}$ and the seasonal minimum of the LMOC.

The seasonal cycle of Lagrangian overturning within the eastern SPG therefore reflects the seasonality of the surface-forced
water mass transformation within the Irminger and Iceland-Rockall Basins north of OSNAP East. The strengthening of the LMOC in summer-autumn corresponds to increasing diapycnal transformation across $\sigma_{LMOC}$, owing to intense surface buoyancy loss along particle trajectories during the ensuing winter. Meanwhile, the weakening of the LMOC through winter-spring reflects a decreasing volume flux into the lower limb as water parcels in the upper limb gain buoyancy along their trajectories during the ensuing summer.





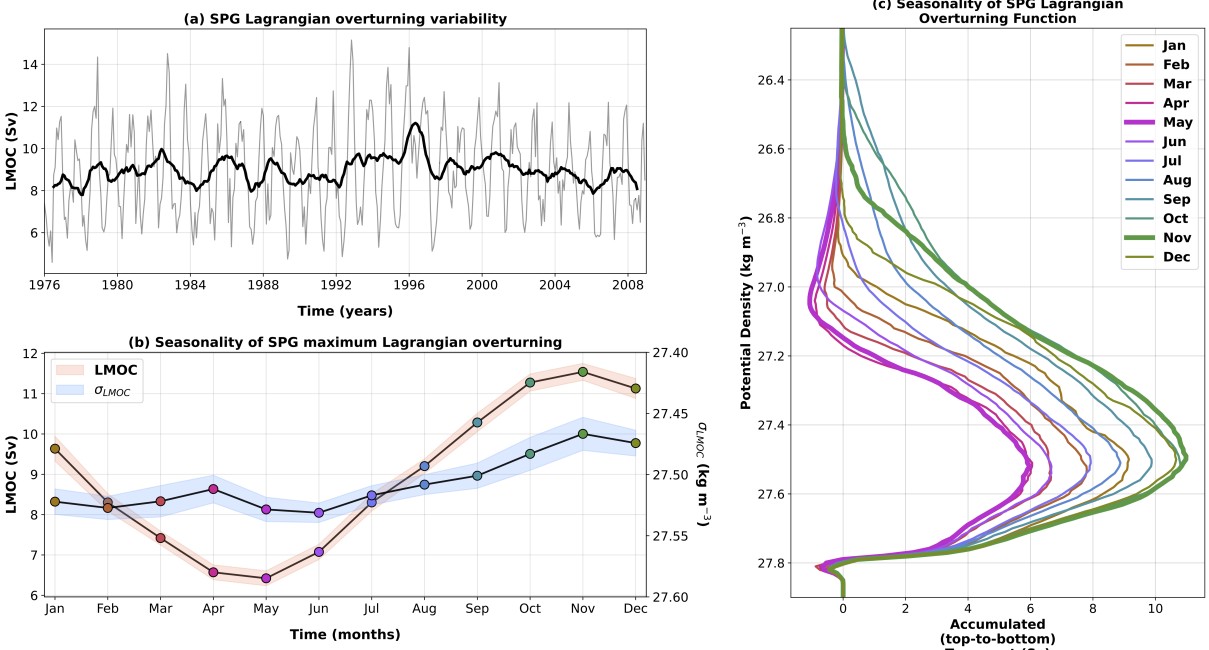

**Figure 3.** Lagrangian overturning variability of the eastern SPG at OSNAP East. (a) Monthly maximum of the Lagrangian Overturning Function in density-space (computed from the subset of particles initialised on the inflow across OSNAP East that recirculate south of the Greenland-Scotland Ridge, grey line) and 1 year annual running mean (bold black line) overlaid for 1976-2008. (b) Mean seasonal cycles of the maximum of the Lagrangian Overturning Function and the isopycnal of maximum Lagrangian overturning ($\sigma_{LMOC}$) computed from monthly composites. Shading represents $\pm 1$ standard error of the monthly estimates. (c) Monthly composites of the Lagrangian Overturning Function calculated in discrete density-space (bin width is 0.01 kg m$^{-3}$); maximum (November) and minimum (May) months in the LMOC seasonal cycle are included in bold.

## 4 Timescales and origins of seasonal Lagrangian overturning

We have demonstrated that variations in the Lagrangian overturning evaluated at OSNAP East are most pronounced on seasonal timescales. However, the magnitude of seasonal variability (SD of seasonal cycle = $\pm 2.1$ Sv) remains small compared with the mean strength of overturning (LMOC = 8.9 Sv), suggesting that the majority of water parcels flowing northwards across OSNAP East in the upper limb are transferred into the lower limb before returning to the section. Since wintertime surface buoyancy loss greatly exceeds summertime buoyancy gain over the eastern SPG (Xu et al., 2018b), the mean strength of the LMOC is governed by the fraction of water parcels which fail to return to OSNAP East before the onset of winter and therefore integrate sufficient surface buoyancy loss to be transferred into the lower limb during their recirculation. In contrast, the seasonality of Lagrangian overturning is determined by rapidly recirculating water parcels whose transformation north of OSNAP East is dependent on the time of year that they arrive at the section. The minimum of the LMOC seasonal cycle (Fig. 3b) is due to the coldest upper limb water parcels arriving at the section in spring, which experience summertime warming



before returning to OSNAP East prior to the onset of wintertime densification. The maximum of the LMOC seasonal cycle, corresponding to the largest volume flux into the lower limb, is due to the warmest upper limb water parcels flowing northwards across the section in autumn, which experience the largest wintertime surface buoyancy loss downstream. This qualitative description of seasonal overturning variability raises two important questions. Firstly, what is the maximum amount of time a

water parcel can spend north of OSNAP East and still contribute to the seasonal cycle of Lagrangian overturning? Secondly, are the water parcels responsible for seasonal variability sourced from distinct inflow regions along OSNAP East when compared with those contributing substantially to the mean state of overturning? To address these questions, we decompose the LOF according to both the time water parcels spend north of OSNAP East (herein referred to as the water parcel recirculation time, $\tau$) and the distance from the East Greenland coast that water parcels flow northward across the section (Fig. 4).

Figure 4a presents the proportion of the mean strength of Lagrangian overturning and amplitude of the seasonal cycle (difference between November-May LMOC monthly composites) accumulated as a function of the time water parcels spend north of OSNAP East ($\tau$). We find that the entire LMOC seasonal cycle and 25% of the mean strength of Lagrangian overturning can be explained by water parcels which spend less than 8.5 months recirculating within the eastern SPG. Interestingly, the absence of any further accumulation of seasonal Lagrangian overturning variability after 8.5 months following northward

inflow across OSNAP East implies that, irrespective of its time of arrival, once a water parcel has experienced wintertime surface buoyancy loss it can no longer imprint onto the seasonal cycle of Lagrangian overturning. Thus, the remaining 75% of the mean LMOC strength is accounted for by water parcels which spend between 8.5 months and 5 years recirculating within the eastern SPG, and experience at least one winter north of OSNAP East. We should note, however, that the accumulation of the mean Lagrangian overturning is not linear over this period; 91% of the volume flux from the upper to the lower limb is

owed to water parcels recirculating in 2 years or less (Fig. 4a).

    To understand how the recirculation time of a water parcel is related to its inflow position along OSNAP East, we next calculate the average recirculation time of water parcels within the eastern SPG as a function of their northward crossing locations along the section. Figure 4b indicates that, on average, the recirculation times of the major upper ocean currents intercepted by OSNAP East are shortest in the upper ocean and increase strongly with depth. This is consistent with observations in the

eastern SPNA, which show that northward transport is surface intensified in the NAC branches (Holliday et al., 2018; Houpert et al., 2018, 2020) and the Irminger Current (de Jong et al., 2020; Fried and de Jong, 2022) since isopycnals in the upper ocean shoal strongly westward. A prominent feature of Figure 4b is the transition from upper limb pathways which contribute to the seasonal cycle of the LMOC (orange, $\tau \leq 8.5$ months) in the Irminger and Central Iceland Basins to longer pathways (blue, $\tau > 8.5$ months), sourced from the central and southern NAC branches, which dominate its mean strength. Figure 4c

quantifies this distinction, highlighting that 74% of the mean strength of the LMOC is due to water parcels originating from the Sub-Arctic Front and the Rockall Trough and Plateau, whereas 96% of the mean seasonal cycle of Lagrangian overturning can be explained by water parcels sourced from the Irminger and Central Iceland Basins ($x \leq 1250$ km). This finding is further supported by the recent results of Li et al. (2021a), who found that changes in the observed velocity and density fields between the East Greenland coast and the Central Iceland Basin can explain 75% of the monthly MOC variance across OSNAP East

between 2014-16.



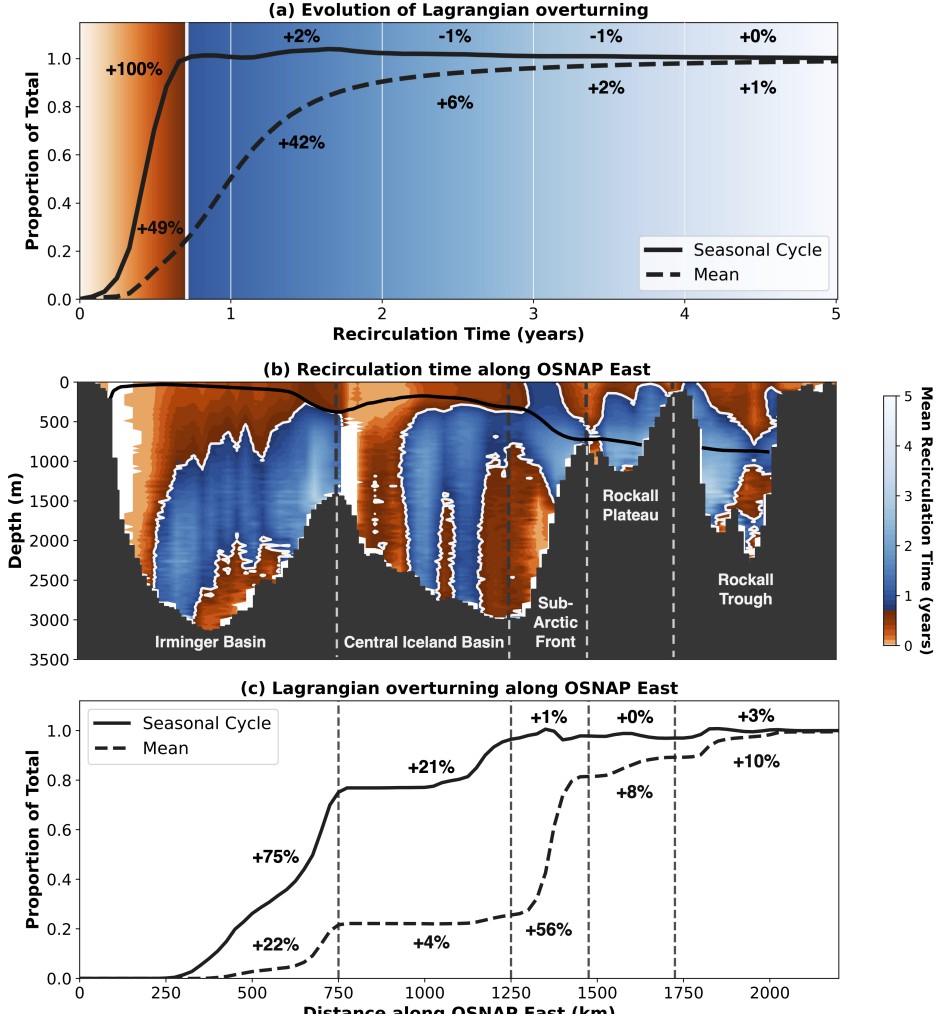

**Figure 4.** Decomposition of the mean strength and seasonality of Lagrangian overturning within the eastern SPG by recirculation time and distance along OSNAP East. (a) Normalised mean strength and seasonality (amplitude of the seasonal cycle calculated as November-May difference between LMOC monthly composites) of the Lagrangian overturning at $\sigma_{LMOC}$ evaluated as a function of the time in which water parcels recirculate back to the OSNAP East section. (b) Mean recirculation time of water parcels initialised on the northward inflow across OSNAP East as a function of their initial position in $x - z$ space, where $x$ is the distance along the OSNAP East section (km) and $z$ is the depth (m). All 9.8 million eastern SPG water parcels initialised across 396 months (1976-2008) are binned according to their initial position, before computing the mean recirculation time for each bin ($\Delta z$ = 25 m, $\Delta x$ = 25 km). White regions coincide with the major southward currents intersected by the OSNAP East array, hence no northward flowing water parcels are initialised here. (c) Normalised mean strength and seasonality of the Lagrangian overturning at $\sigma_{LMOC}$ accumulated as function of the distance from the East Greenland coast that water parcels flow northward across OSNAP East.



To summarise, we have identified a threshold recirculation time of 8.5 months which governs whether water parcels will contribute substantially to the mean strength of the LMOC or determine its seasonal signal at OSNAP East. Seasonal overturning variability is sourced from the upper Irminger and Central Iceland Basins, where the rapid recirculation of water parcels in less than 8.5 months yields along-stream transformations which are dependent on their time of arrival at OSNAP East. Conversely, the mean strength of the LMOC is determined by water parcels, originating from the central and southern NAC branches, whose longer recirculation time ($\tau > 8.5$ months) guarantees their transformation into the LMOC lower limb through intense wintertime buoyancy loss.

## 5 Seasonal Lagrangian overturning pathways within the eastern SPG

### 5.1 Pathways of seasonal overturning variability

Following Tooth et al. (2022), we classify water parcels according to four cyclonic circulation pathways defined between OSNAP East and the Greenland-Scotland Ridge. The majority of water parcels recirculate exclusively within the Iceland-Rockall and Irminger Basins and are referred to as the ***IcRo-IcRo*** and ***Irm-Irm*** pathways, respectively (see Fig. 5a for regional definitions). The remaining water parcels flow westward across the Reykjanes Ridge before crossing OSNAP East in the East Greenland Current, which flows southward along the western boundary of the Irminger Basin. These water parcels are subdivided into two pathways according to their inflow location along the section: the ***Ic-RR-Irm*** pathway comprises water parcels sourced from the central NAC branch positioned along the Sub-Arctic Front, while the ***Ro-RR-Irm*** pathway originates from the southern NAC branch feeding both the Rockall Trough and the Rockall Plateau (Fig. 5b).

Concordant with our earlier analysis, Figure 5 shows a clear distinction between the circulation pathways responsible for the seasonality of the LMOC and those governing its mean state. We find that the two pathways crossing the Reykjanes Ridge north of OSNAP East account for 70% of the mean strength of the Lagrangian overturning within the eastern SPG yet exhibit negligible variability on seasonal timescales (Fig. 5c). This is because the recirculation times of water parcels advected across the ridge north of OSNAP East consistently exceed the critical 8.5 month threshold required to be irreversibly transferred into the lower limb of the LMOC. Water parcels advected along the ***Ic-RR-Irm*** pathway typically experience 1.1 years (Fig. 5d) of along-stream surface buoyancy loss in order to transfer $4.8 \pm 1.0$ Sv of water into the lower limb. Water parcels flowing northward along the slower ***Ro-RR-Irm*** pathway ($\bar{\tau}$ = 1.7 years) form $1.4 \pm 0.4$ Sv of dense Icelandic Slope Water (Van Aken and De Boer, 1995; Read, 2000) by entraining Iceland Scotland Overflow Water (ISOW) in the vicinity of the Faroe Bank Channel (Fig. 5a).

Figure 5c shows that water parcels recirculating exclusively within the Irminger Basin (***Irm-Irm***) explain 75% of the amplitude of the LMOC seasonal cycle at OSNAP East (Fig. 4). However, our use of a single ***Irm-Irm*** pathway obscures two separate pathways that circulate cyclonically within the Irminger Sea, namely the Irminger Current ***IC*** and the Irminger Gyre ***IG***. Therefore, to isolate the Lagrangian overturning occurring along each of these pathways, we define the boundary between the ***IG***, recirculating within the basin interior, and the ***IC***, positioned on the western flank of the Reykjanes Ridge, to be 500 km from Cape Farewell, following Våge et al. (2011). We find that the northward inflows to the ***IG*** ($x \leq 500$ km) make a



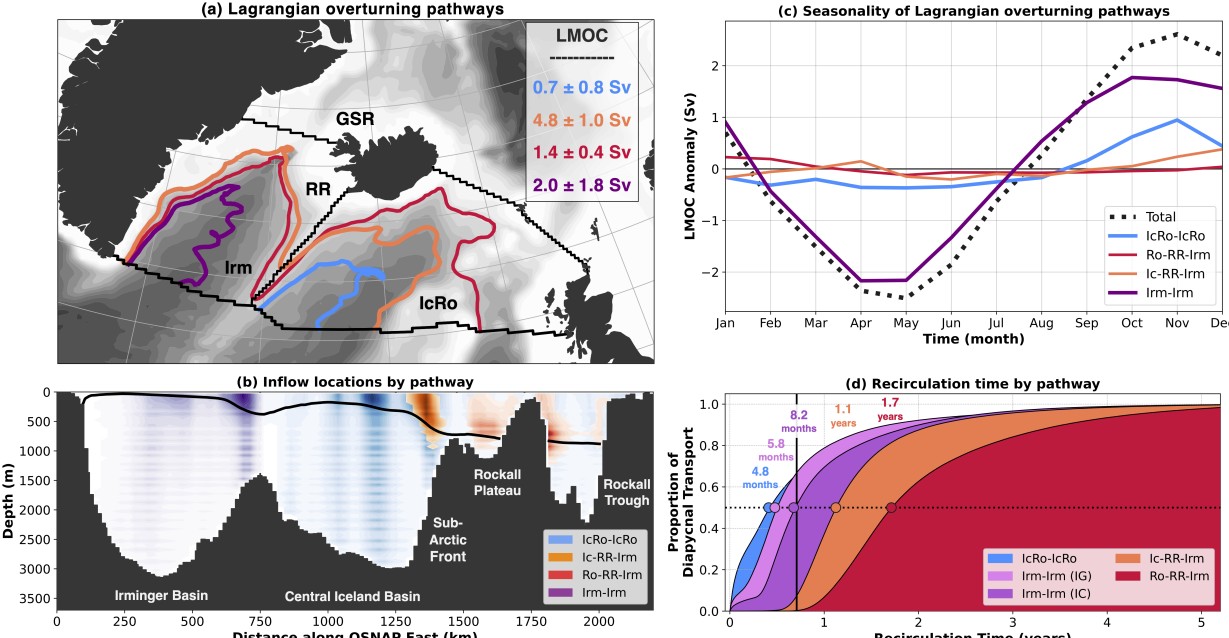

**Figure 5.** Decomposition of the seasonal cycle of Lagrangian overturning within the eastern SPG by circulation pathway north of OSNAP East. (a) Example water parcel trajectories illustrating each of the four circulation pathways within the eastern SPG and their respective contributions to the mean strength of Lagrangian overturning at $\sigma_{LMOC}$. (b) Inflow locations where water parcels are advected northward across OSNAP East classified by pathway. Water parcel volume transports for each pathway are averaged in discrete $x$ - $z$ space ($\Delta z = 25$ m, $\Delta x = 25$ km) using all 396 initialisation months before normalising by the maximum mean transport recorded across all bins (darker shading corresponds to the strongest northward transport for each pathway). (c) Seasonal cycle of the Lagrangian overturning at $\sigma_{LMOC}$ decomposed by circulation pathway north of OSNAP East. For clarity, variations in the LMOC are presented as anomalies about the time-mean overturning determined from monthly composites of each pathway. (d) The proportion of the total diapycnal transport ($|\Delta\sigma| > 0.01$ kg m$^{-3}$, where $\Delta\sigma$ is the net change in potential density between northward and southward crossing of OSNAP East) to have successfully recirculated back to OSNAP East as a function of the time elapsed following northward flow across the section. Coloured circles correspond to the median recirculation time of each circulation pathway.

disproportionately large contribution to the seasonality of Lagrangian overturning (26%) compared with their contribution to
its mean strength (3%). Water parcels advected along the path of the **IC** (500 km $< x \leq 750$ km) explain almost half (49%) of the seasonal cycle of Lagrangian overturning whilst also accounting for approximately a fifth (19%) of the mean volume flux across $\sigma_{LMOC}$. The larger overturning associated with the **IC** is explained by Figure 5d, which shows that water parcels typically spend 8.2 months recirculating within the **IC** compared with only 5.8 months within the **IG**. Since the median recirculation time of the **IC** pathway is comparable to the 8.5-month threshold required to avoid wintertime densification north of





OSNAP East, it follows that approximately half of all water parcels advected along the boundary current of the Irminger Sea will be transformed across $\sigma_{LMOC}$.

Although watermass transformation along the **IcRo-IcRo** pathway accounts for only a quarter of the LMOC seasonal cycle at OSNAP East, Figure 5c highlights its critical role in establishing the timing of the November maximum in the seasonal cycle of Lagrangian overturning. Notably, it is the rapid recirculation of water parcels from the northern NAC branch to the

East Reykjanes Ridge Current (ERRC) within 4.8 months that dominates seasonal overturning variability along this pathway. Figure 5c shows that the largest volume flux into the lower limb of the LMOC occurs in November when **IcRo-IcRo** water parcels can integrate the greatest surface buoyancy loss throughout the ensuing winter spent north of OSNAP East. This is consistent with the results of de Boisséson et al. (2010), who calculated the strongest net heat loss over the Iceland Basin between November and February. In contrast, a weak negative volume flux (directed from the lower to the upper limb) is

found when **IcRo-IcRo** water parcels flow northward between January-August. This corresponds to the obduction of cold, fresh subpolar water across the isopycnal of maximum Lagrangian overturning within the Central Iceland Basin. Brambilla et al. (2008) attributed such negative diapycnal transformation to divergence along the northern NAC branch, which induces upwelling and hence mixing with warm, saline subpolar mode waters in the mixed layer.

## 5.2 Transformation along seasonal overturning pathways

To understand how seasonal Lagrangian overturning variability results from diapycnal transformation along water parcel trajectories, we next calculate the net potential density change of water parcels between northward and southward crossings of OSNAP East, $\Delta\sigma$, as a function of their inflow location along the section. Figure 6a-b highlights the striking disparity between the net diapycnal transformation of water parcels flowing northward in the Irminger and Central Iceland Basins during May (lightening) and November (densification). We find that the largest negative diapycnal transformations occur along water par-

cel trajectories sourced from the upper 100m of the **IG** and the Central Iceland Basin (**IcRo-IcRo**) in May, corresponding to the minimum of the LMOC seasonal cycle. When the LMOC reaches its maximum in November, water parcels originating from the upper 250 m of the **IC** undergo strong positive diapycnal transformation, consistent with the expansion of the upper limb through summertime restratification. In contrast, water parcels sourced from the upper kilometer of central and southern branches of the NAC exhibit densification whatever time of year they flow north since they are guaranteed to experience at

least one winter north of OSNAP East during their recirculation.

Since water parcels recirculating cyclonically along the boundary current of the Irminger Sea typically spend an additional 2.4 months north of OSNAP East compared with those circulating in the interior of the Irminger and Central Iceland Basins (Fig. 5), we next explore how the character of seasonal water mass transformation differs along boundary and interior pathways within the eastern SPG. We focus our analysis on the water parcels circulating in the upper 250 m of the Irminger Sea, since

these collectively account for three quarters of the seasonal cycle of Lagrangian overturning at OSNAP East. To compare the boundary and interior modes of seasonal overturning variability, Figure 6c-d presents the transport-weighted mean potential density of upper **IG** (pink box in Fig. 6a-b) and upper **IC** (purple box in Fig. 6a-b) water parcels on their northward (IG/IC inflow) and subsequent southward (EGC outflow) crossings of the OSNAP East section. We find the densest water flowing




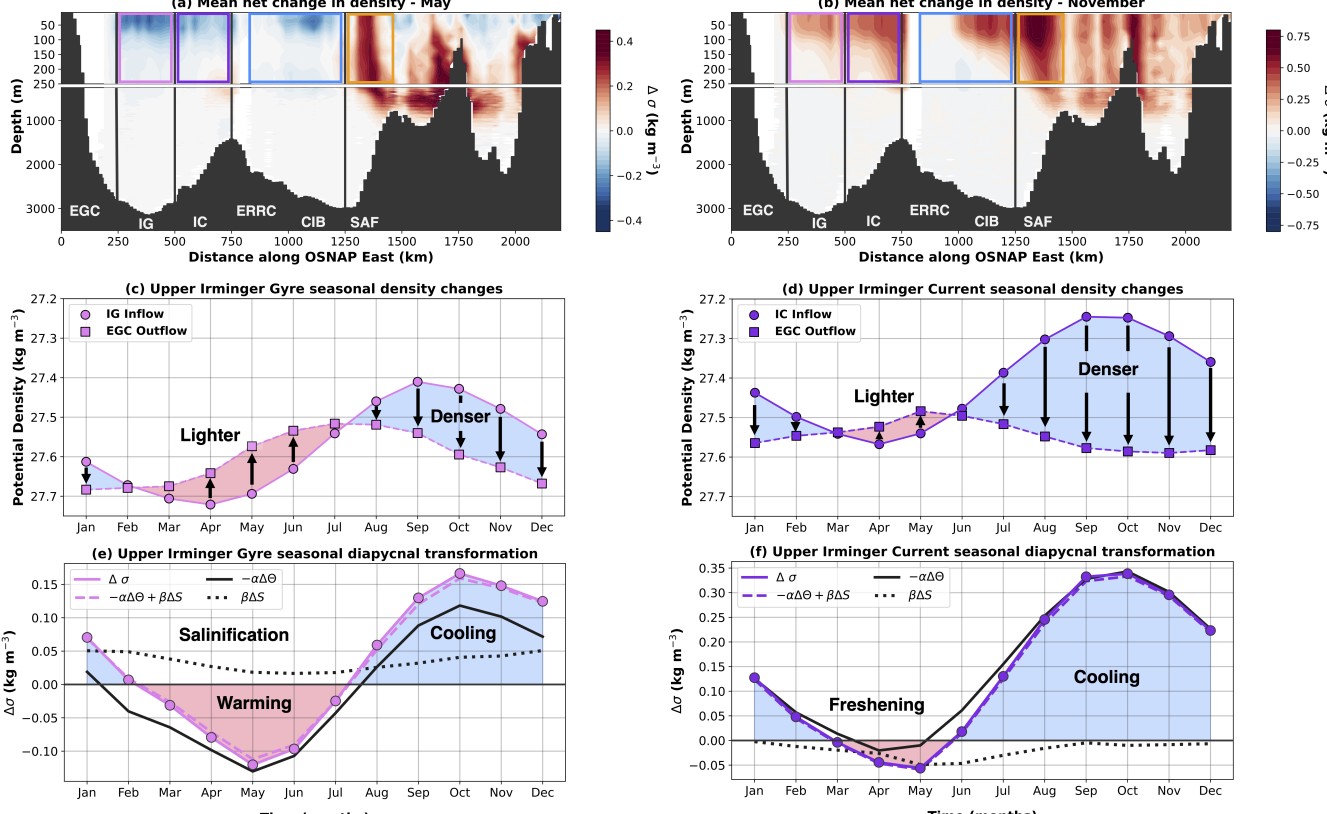

**Figure 6.** (a) Mean net change in potential density of water parcels flowing northward across OSNAP East in May as a function of their inflow location along OSNAP East. The net change in potential density, $\Delta\sigma$ (kg m$^{-3}$), between northward and southward crossings of OSNAP East is computed for each recirculating water parcel before averaging in discrete $x - z$ space ($\Delta z = 25$ m, $x = 25$ km). (b) Same as (a) for water parcels flowing northward in November. Vertical solid lines correspond to the geographical limits of the major currents intercepted along the section. Boxes included in the upper 250 m define the upper **IG** (pink), **IC** (purple), **IcRo-IcRo** (blue) and **Ic-RR-Irm** (orange) pathways. (c, d) Mean seasonal cycles of transport-weighted mean potential density on inflow (circular markers) and outflow (square markers) crossing of OSNAP East for the upper (c) **IG** and (d) **IC** pathways. (e, f) Mean seasonal cycle of transport-weighted mean net change in potential density and contributions from diathermal ($\Delta\theta$, black solid line) and diahaline ($\Delta S$, black dotted line) components for the upper (e) **IG** and (f) **IC** pathways.

northward in both the upper **IG** and upper **IC** in April, consistent with the occurrence of the deepest mixed layers and the outcropping of $\sigma_{LMOC}$ following wintertime deep convection (de Jong et al., 2012; de Jong and de Steur, 2016; Piron et al., 2016). The lightest water parcels arrive in the upper 250 m of the Irminger Sea during September when upper ocean stratification is strongest following summertime heating. Despite the similar seasonality in their mean potential density on northward inflow, upper **IG** water parcels are consistently denser than those initialised in the upper **IC** (Fig. 6c-d) owing to their convective origins in the Labrador Sea interior (Lavender et al., 2000; Chafik et al., 2022). Moreover, upper **IG** water parcels exhibit a





much narrower potential density range (27.52 - 27.68 kg m$^{-3}$) on southward outflow compared with those recirculating along the upper **IC** pathway (27.48 - 27.59 kg m$^{-3}$). This is because the longer recirculation time of the upper **IC** pathway ($\bar{\tau}$ = 6.2 months) allows water parcels to undergo greater surface buoyancy loss along the boundary current, thereby damping the seasonality of their water mass properties on inflow. The additional time upper **IC** water parcels spend north of OSNAP East is also reflected by the shorter 2-month window (April-May in Fig. 6d) during which negative diapycnal transformation can occur, compared with the 5-month window (March-July in Fig. 6c) for the rapidly recirculating upper **IG** pathway ($\bar{\tau}$ = 3.8 months).

To determine the relative importance of temperature and salinity changes along boundary and interior pathways, we further decompose the net diapycnal transformation ($\Delta\sigma$) north of OSNAP East into diathermal and diahaline components. We approximate $\Delta\sigma$ using a linearised form of the equation of state following the integral approach of Tamsitt et al. (2018):

$$\Delta\sigma \approx -\alpha(\bar{\theta}, \bar{S})\Delta\theta + \beta(\bar{\theta}, \bar{S})\Delta S, \tag{3}$$

where $\theta$ denotes potential temperature, $S$ denotes salinity, $\alpha$ represents the thermal expansion coefficient, and $\beta$ represents the haline contraction coefficient. The values of $\bar{\theta}$ and $\bar{S}$ correspond to the average potential temperature and salinity of a water parcel on its northward and southward crossings of OSNAP East.

Figures 6e-f shows close agreement between the seasonal cycles of net diapycnal transformation (pink/purple solid line) along the upper **IG/IC** pathways and its reconstruction, given by the sum of diathermal ($-\alpha\Delta\theta$) and diahaline ($\beta\Delta S$) components (pink/purple dashed line). Consistent with the strong seasonality of surface heat fluxes over the Irminger Basin (de Jong and de Steur, 2016; Piron et al., 2016), we find that the seasonal cycle of diapycnal transformation along both boundary and interior pathways is dominated by diathermal transformation along water parcel trajectories. In contrast, along-stream diahaline transformations play opposing roles along the upper **IG** and upper **IC** pathways. Figure 6e shows that water parcels rapidly recirculating in the interior of the Irminger Basin experience year-round salinification north of OSNAP East. Salinification is strongest when water parcels are initialised during winter, suggesting that lateral mixing between the cold, fresh interior and warmer and saltier SPMWs advected along the boundary current is enhanced during the ensuing months. This proposition is supported by the recent observations of de Jong et al. (2020), who found that eddy kinetic energy is largest near the surface of the Irminger Current between January and April. For upper **IG** water parcels flowing northward across OSNAP East from August-January, mixing induced salinification acts to augment along-stream densification due to intense surface heat loss over the basin interior. Although the predominant influence of boundary-interior exchange is year-round positive diahaline transformation, we should also note that the diathermal consequence of mixing is that summertime warming of **IG** water parcels exceeds their wintertime cooling (Fig. 6e).

Diahaline transformation along the upper **IC** pathway is characterised by the freshening of water parcels flowing northward across OSNAP East between February-August. Interestingly, Figure 6f shows that it is this substantial freshening, rather than summertime heating, which is responsible for the negative diapycnal transformation along water parcel trajectories initiated in late spring - early summer. The largest freshening along the boundary current is associated with water parcels which flow



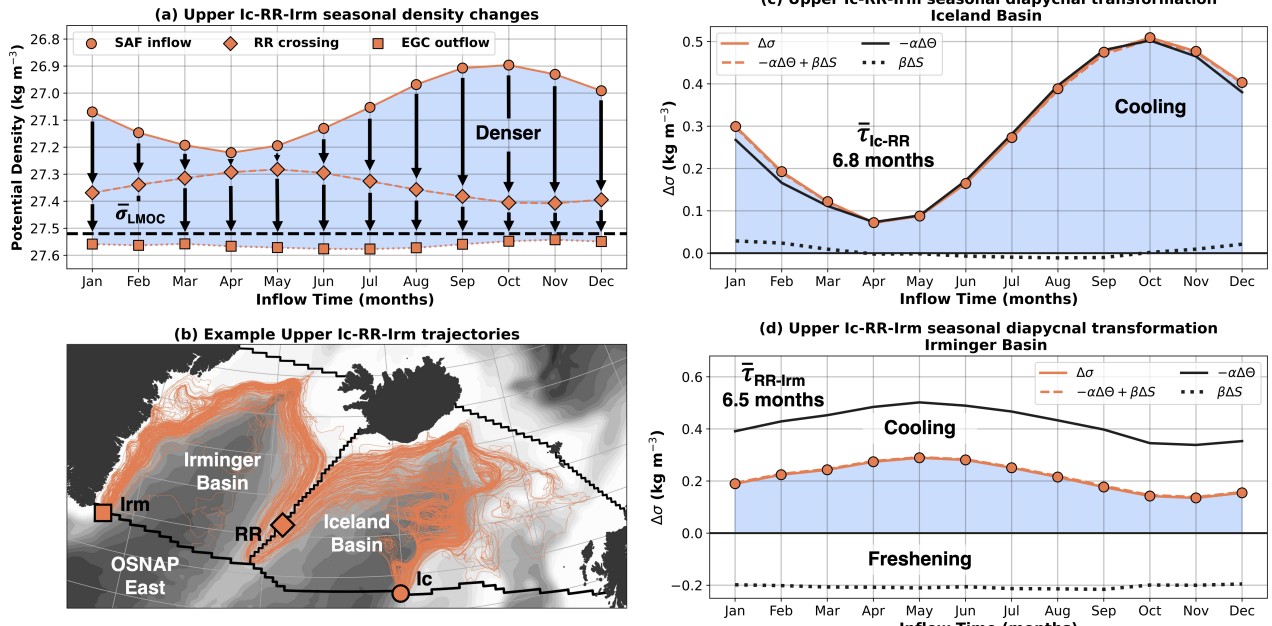

**Figure 7.** (a) Mean seasonal cycle of transport-weighted mean potential density on SAF inflow (circular markers), RR crossing (rhombus markers) and EGC outflow (square markers) for water parcels flowing northward across OSNAP East in the upper 250 m of the ***Ic-RR-Irm*** pathway. Note that the transport-weighted mean potential densities on crossing the RR and returning to OSNAP East via the EGC are plotted at the month when water parcels originally flowed northward across OSNAP East. (b) Example water parcel trajectories illustrating the upper ***Ic-RR-Irm*** circulation pathway within the eastern SPG. (c, d) Mean seasonal cycles of transport-weighted mean net change in potential density, $\Delta\sigma$, and contributions from diathermal ($\Delta\theta$, black solid line) and diahaline ($\Delta S$, black dotted line) components divided into two successive diapycnal transformations in the Iceland (c, Ic-RR) and Irminger Basins (d, RR-Irm)

northward in the ***IC*** during May-June and return southward across OSNAP East via the EGC in November-December. This is consistent with observations, which show that the largest freshwater transport of the East Greenland Coastal Current (EGCC)
occurs during autumn (Daniault et al., 2011; Le Bras et al., 2018) following the summertime accumulation of meltwater on the shelf. Moreover, observations indicate that the strongest mixing between cold, fresh EGCC water and comparatively warmer and saltier EGC water occurs in winter (Le Bras et al., 2018), concurrent with the return of the freshest upper ***IC*** water parcels across OSNAP East. Notably, Figure 6f indicates that upper ***IC*** trajectories flowing northward across the section between September-January (returning southward between March-June) experience negligible diahaline transformation alongstream,
since freshwater transport in the EGCC is weakest during spring (Le Bras et al., 2018; Lin et al., 2018). Thus, the largest seasonal diapycnal transformations along the boundary current of the Irminger Sea are dominated by surface heat loss.

Since water flowing northward in the upper ***IC*** is sourced from relatively warm and saline SPMWs advected along the northern branch of the NAC (McCartney and Talley, 1982; Brambilla and Talley, 2008; Daniault et al., 2016), it is interesting to compare and contrast the character of seasonal water mass transformation along the upper ***IC*** pathway with the dominant



*Ic-RR-Irm* overturning pathway within the eastern SPG. Figure 7 presents the seasonal density changes and their associated diathermal and diahaline components for water parcels crossing the OSNAP East section in the upper 250 m and following the *Ic-RR-Irm* pathway. Given that the *Ic-RR-Irm* pathway flows from the Sub-Arctic Front to the EGC via the Reykjanes Ridge, we choose to decompose the net diapycnal transformation north of OSNAP East into two successive transformations taking place in the Iceland and Irminger Basins (Fig. 7c-d). Figure 7a shows that upper *Ic-RR-Irm* water parcels experience sufficient net diapycnal transformation north of OSNAP East to be transferred into the lower limb of the LMOC, irrespective of their water mass properties on inflow. This is because water parcels typically recirculate on interannual timescales ($\bar{\tau}$ = 1.1 years) and are therefore guaranteed to experience at least one winter within the eastern SPG. For the lightest water parcels flowing northward in autumn, this intense wintertime heat loss manifests as a large diathermal transformation (equivalent to ∼0.5 kg m$^{-3}$) during their initial 6.8 months spent within the Iceland Basin (Fig. 7c). In contrast, the densest water parcels arriving at OSNAP East in spring experience wintertime diathermal transformation (equivalent to ∼0.5 kg m$^{-3}$) during their final 6.5 months spent in the Irminger Basin (Fig. 7d). Thus, we find that wintertime surface buoyancy loss removes all seasonal thermohaline variability flowing northwards into the eastern SPG. Meanwhile, the transit times of upper *Ic-RR-Irm* water parcels through the Iceland and Irminger Basins account for the 7-month phase shift in the seasonal cycle in density as it decays downstream (Fig. 7a).

In addition to surface-forced diapycnal transformations, the remarkably consistent density of the upper *Ic-RR-Irm* pathway on returning southward across OSNAP East in the EGC (27.55 ± 0.01 kg m$^{-3}$) underscores the importance of interior mixing along water parcel trajectories. This is because interior mixing is associated with the convergence of water parcels in T-S space, whereas transformation by surface fluxes leads to diverging water mass properties (Groeskamp et al., 2014; Mackay et al., 2020). The strong density compensation between year-round cooling and freshening along the upper *Ic-RR-Irm* pathway in the Irminger Sea (Figure 7d) is indicative of a constant background mixing with fresher Arctic-origin waters transported in the EGCC. A closer examination shows that this mixing (freshening) along water parcel trajectories is concentrated near the Kangerdlugssuaq Trough in the northern Irminger Basin, where observations have found substantial freshwater transports directed offshore from the East Greenland Shelf (Sutherland and Pickart, 2008; Sutherland et al., 2009; Foukal et al., 2020). We therefore propose that mixing between Arctic and Atlantic water masses along the western boundary of the Irminger Sea plays an integral role in maintaining the stable composition of the lower limb on seasonal timescales (see Fig. 3b where $\sigma_{LMOC}$ = 27.51±0.02 kg m$^{-3}$).

In summary, we have identified two dominant modes of seasonal Lagrangian overturning variability taking place in the upper 250 m of the eastern SPG. Approximately half of the seasonal cycle of Lagrangian overturning at OSNAP East originates from the boundary current encircling the Irminger Sea, while the remainder is sourced equally from water parcels recirculating in the interior of the Irminger and Central Iceland Basins. Seasonal diapycnal transformation along both the boundary and interior pathways is dominated by the diathermal component resulting from air-sea interaction north of OSNAP East. Although diahaline transformations are of secondary importance to boundary and interior densification, they highlight the important and contrasting roles of lateral mixing between water masses along-stream. In the upper Irminger Gyre, the year-round salinification of water parcels owing to boundary-interior exchange acts to reinforce their wintertime densification. Meanwhile, water parcels



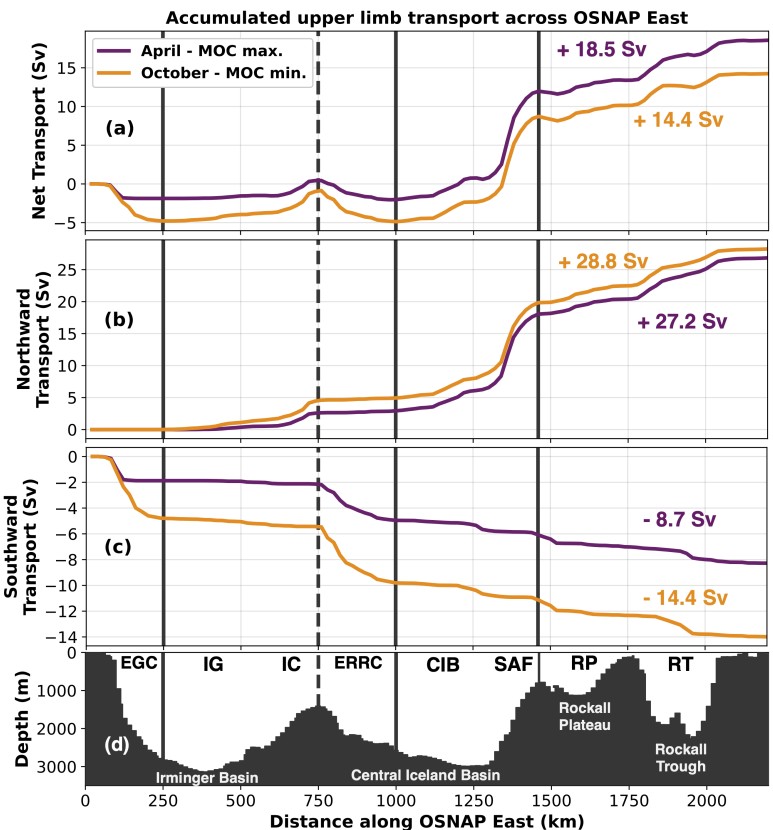

**Figure 8.** Mean Eulerian volume transport in the upper limb (i.e. above $\sigma_{MOC}$) of the MOC accumulated with distance eastward along OSNAP East. The net upper limb volume transport (a) is decomposed into its respective northward (b) and southward components (c). The model bathymetry along the OSNAP East array is presented in (d). Vertical lines partition the volume transport across the section into five geographical regions corresponding to the major currents intercepted by the array: East Greenland Current (EGC), Irminger Gyre and Irminger Current (IG & IC), East Reykjanes Ridge Current (ERRC), the Central Iceland Basin and Sub-Arctic Front (CIB & SAF), and the Rockall Plateau and Rockall Trough (RP & RT).

flowing northward across OSNAP East in the boundary current in spring exhibit substantial freshening along-stream due to enhanced mixing with Arctic-origin waters in the EGCC during the ensuing autumn. The additional 2.4 months required for water parcels to be advected along the Irminger Current also reduces the variability of water mass properties on southward outflow compared with those rapidly recirculating in the Irminger Gyre. When advective timescales are increased further, as found along our dominant overturning pathway (**_Ic-RR-Irm_**), intense wintertime heat loss north of OSNAP East acts to damp

the seasonality of water mass properties on northward inflow, thus forming lower limb water of remarkably consistent density.





## 6   Mechanisms of seasonal Eulerian overturning variability

We conclude our analysis by using Lagrangian water parcel trajectories to cast new light on the mechanisms responsible for seasonal Eulerian overturning variability. We begin by decomposing the net volume flux in the MOC upper limb, determined directly from the simulated velocity and potential density fields along OSNAP East, into its constituent northward and south-

ward components during April and October, corresponding to the extrema of the MOC seasonal cycle in Figure 2b. Figure 8 indicates that, in spite of the year-round net northward transport in the upper limb, the seasonal cycle of the MOC results from changes in the southward transport above $\sigma_{MOC}$ at OSNAP East. This is highlighted in Figure 8c, which shows that the total southward transport in the upper limb increases significantly from -8.7 Sv in April, when the Eulerian overturning reaches its seasonal maximum (18.5 Sv), to -14.4 Sv when the MOC seasonal minimum (14.4 Sv) occurs in October.

**Table 1.** Mean Eulerian net volume transport (Sv) in the upper limb of the MOC (i.e., above $\sigma_{MOC}$) for the major currents along the OSNAP East array during April and October. The major currents are defined geographically as follows: East Greenland Current (EGC, 0 km $< x \leq$ 250 km), Irminger Gyre and Irminger Current (IG & IC, 250 km $< x \leq$ 750 km), East Reykjanes Ridge Current (ERRC, 750 km $< x \leq$ 1000 km), Central Iceland Basin and Sub-Arctic Front (CIB & SAF, 1000 km $< x \leq$ 1450 km), and Rockall Plateau and Rockall Trough (RP & RT, 1450 km $< x \leq$ 2300 km). Note $x$ corresponds to the distance from Cape Farewell on the east Greenland coast.

| | EGC | IG & IC | ERRC | CIB & SAF | RT & RP | Total |
|---|---|---|---|---|---|---|
| **Net Upper Limb Volume Transport (Sv)** | | | | | | |
| April | -2.1 | 2.4 | -2.5 | 14.2 | 6.5 | 18.5 |
| October | -4.9 | 4.2 | -4.2 | 13.9 | 5.4 | 14.4 |

Further decomposition of the seasonal upper limb transport according to the major currents crossing the OSNAP East array (Table 1) indicates that the seasonal minimum of the MOC results from the combination of a 2.8 Sv strengthening of the EGC southward transport above $\sigma_{MOC}$ and a 1.1 Sv weakening of the southern NAC branch feeding the Rockall Trough and Plateau. This agrees with the recent results of Wang et al. (2021), who demonstrated that variations in the southward transport along the western boundary of the Irminger Sea play a prominent role in modulating the seasonal cycle of overturning at OSNAP East.

Moreover, Wang et al. (2021) showed that the seasonality of the EGC upper limb transport is principally explained by seasonal density changes in the upper Irminger Sea projecting onto the mean barotropic transport of the western boundary current. To explore this further, Figure 9 presents the mean potential density field along OSNAP East in April (MOC maximum) and October (MOC minimum) and the corresponding location of $\sigma_{MOC}$. In April, we find that the erosion of stratification, owing to intense wintertime heat loss, permits deep convective mixing in the Irminger Sea interior (de Jong et al., 2012; de Jong

and de Steur, 2016; Piron et al., 2016), such that $\sigma_{MOC}$ (27.57 kg m$^{-3}$ in April) outcrops at the surface (Fig. 9a). As a consequence, the total northward transport above $\sigma_{MOC}$ along OSNAP East is reduced, since a substantial fraction of the water flowing northward in the Irminger Sea does so in the lower limb of the MOC (de Jong et al., 2020). Along the western





boundary, the isopycnal of maximum overturning slopes steeply with distance offshore during spring, resulting in a weak upper

limb transport in the EGC (-2.1 Sv, Table 1). Thus, since less southward transport is available to compensate for the stronger

northward transport above $\sigma_{MOC}$ in the NAC (20.7 Sv, Table 1), the MOC reaches a seasonal maximum. A contrasting picture

emerges in October, when Figure 9b shows that surface heating through summer has restored the stratification in the upper

Irminger Sea. This is in agreement with the recent observations of de Jong et al. (2020), who found the lowest monthly mean

density along OSNAP East during October. Since the depth of $\sigma_{MOC}$ in Figure 9b exceeds 100 m throughout the entire

Irminger Basin, there is hence a larger northward upper limb transport in the Irminger Gyre and the Irminger Current (4.2 Sv).

However, this is more than compensated for by the vertical migration of the isopycnal of maximum overturning offshore of the

East Greenland shelfbreak, which enables water flowing southward in the upper 200 m of the EGC to be included within the

lighter MOC upper limb ($\sigma_{MOC} = 27.50\ \mathrm{kg\ m^{-3}}$ in October) as recently observed by Le Bras et al. (2020). This, in conjunction

with the weaker transport of warm, saline water flowing northward in the NAC (a change of -1.4 Sv), is therefore responsible

for the MOC seasonal minimum in October.

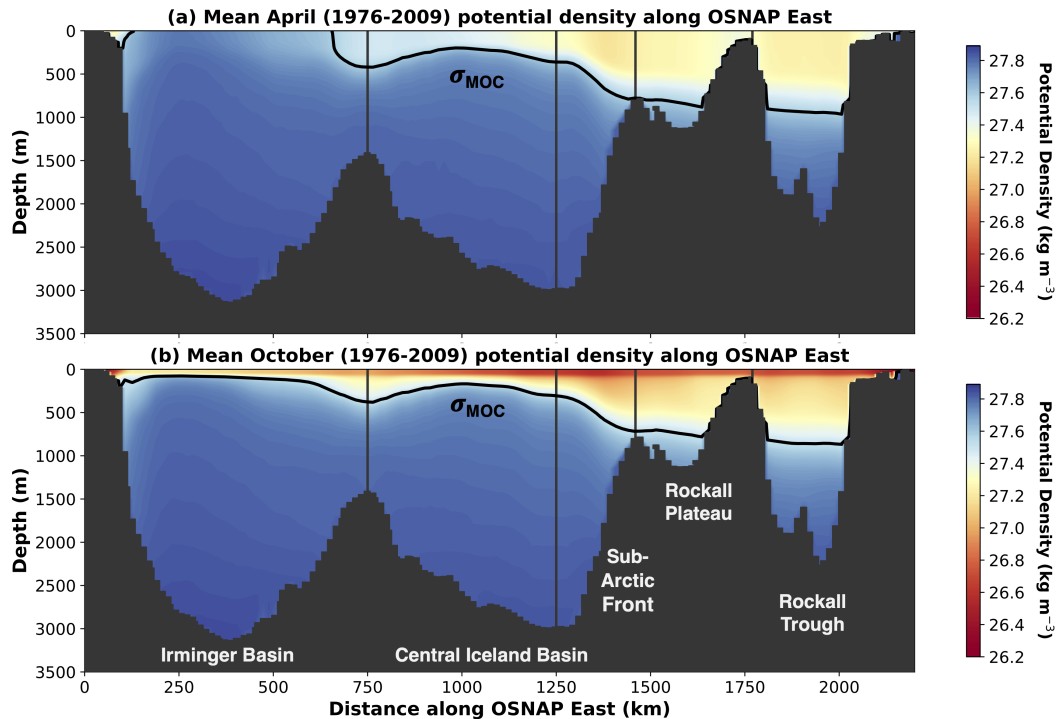

**Figure 9.** Mean potential density along OSNAP East in (a) April and (b) October overlaid by the corresponding mean isopycnal of maximum
Eulerian overturning (April: $\sigma_{MOC} = 27.57\ \mathrm{kg\ m^{-3}}$, October: $\sigma_{MOC} = 27.50\ \mathrm{kg\ m^{-3}}$), denoting the seasonal interface between the upper
and lower limbs of the MOC.

In contrast with the conclusion of Wang et al. (2021), we do not find that the seasonal cycle of the MOC can be accounted for

by changes in the density structure of the western Irminger Basin alone. Rather, we propose that variations in the velocity field





of the EGC act in conjunction with seasonal migrations of $\sigma_{MOC}$ to drive the seasonality of Eulerian overturning at OSNAP East. To demonstrate this, we return to the water parcels recirculating exclusively within the upper 250 m of the Irminger Basin (upper ***Irm-Irm***), where we find a strong seasonal signal in the median amount of time water parcels spend north of OSNAP

East (Fig. 10). Figure 10a shows that recirculation times along both the upper ***IC*** and ***IG*** pathways are longest when water parcels flow northward across OSNAP East during winter and are shortest when they cross the section during summer. For the upper ***IC*** pathway, this amounts to water parcels typically spending 8 months circulating along the boundary current following initialisation in February compared with only 6 months when initialised in August. While this gives us a useful indication of the seasonality in the large-scale circulation north of OSNAP East, it is the transition in upper ***Irm-Irm*** recirculation times through

spring (Fig. 10a) which is the crucial driver of seasonal MOC variability. This is because the decrease in the recirculation times of water parcels crossing OSNAP East northwards in the upper ***IC*** between February-May produces a convergence of water parcels flowing southward in the EGC during autumn. The strongest convergence occurs in October, consistent with the observed intensification of the upper EGC at OSNAP East during autumn (Le Bras et al., 2018; Pacini et al., 2020), and amounts to a 1.0 Sv negative anomaly in the full-depth transport of the EGC. While the magnitude of this transport anomaly

remains small compared with the typical magnitude of the EGC in this simulation (31.0 Sv), its surface-intensified nature has important consequences for the strength of the MOC at OSNAP East. Figure 10a shows that the convergence of upper ***Irm-Irm*** water parcels within the EGC occurs almost exclusively within the upper limb of the MOC. Thus, by acting in concert with the deepening of $\sigma_{MOC}$ during late-summer and autumn (Fig. 9b), the convergence of upper ***Irm-Irm*** water parcels within the boundary current can explain 2.6 Sv of the 2.8 Sv increase in the upper limb EGC southward transport between April and

October, accounting for almost two-thirds of the amplitude of the MOC seasonal cycle at OSNAP East (4.2 Sv).

Previous studies have highlighted the close relationship between seasonal variations in the large-scale circulation of the Irminger Sea and wind-stress forcing acting over the basin (Daniault et al., 2011; Le Bras et al., 2018). To determine whether local wind-stress forcing can account for the seasonality of water parcel recirculation times in the upper Irminger Sea, we next compare the character of upper ***IC*** trajectories flowing northward across OSNAP East in February and August, corresponding

to the longest and shortest recirculation times, respectively. Figure 10b shows that the longer median recirculation times of upper ***IC*** water parcels flowing northward across OSNAP East in February is due to their slower advection along the boundary current through spring-summer. The weakening of the boundary current is explained by the springtime spin-down of the SPG circulation owing to the decrease in wind-stress curl acting over the Irminger Sea (Daniault et al., 2011). In addition to their slower recirculation times along the boundary current, water parcels flowing northward in February are more likely to be

entrained into slower circulation pathways in the basin interior, resulting in a longer tail in the distribution of recirculation times in Figure 10c. The shorter recirculation times exhibited by upper ***IC*** water parcels flowing northward in August results from the autumn-wintertime spin-up of the SPG during their recirculation north of OSNAP East. Figure 10b shows that water parcels flowing northward in August experience especially fast advection within the EGC, where the strongest north-easterly winds act along-stream. This is consistent with the study of Le Bras et al. (2018), who found that seasonal variations in the

wind-stress curl, and by extension the EGC transport, are largely determined by changes in the local wind stress field acting along the East Greenland coast.



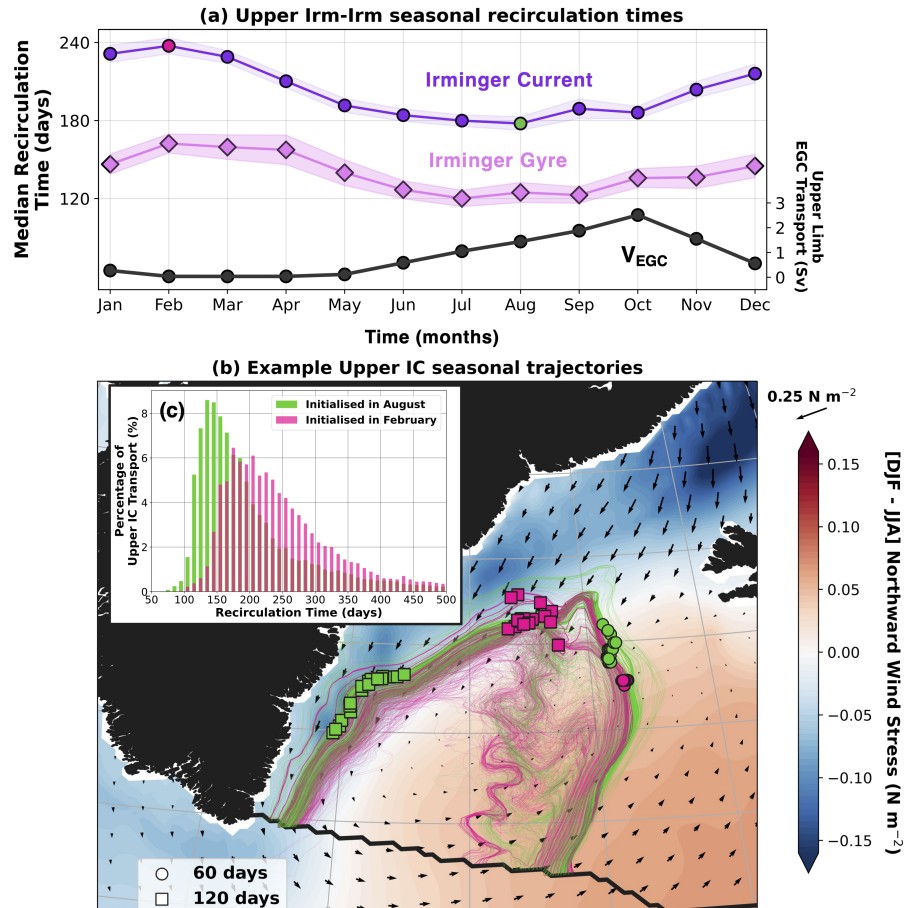

**Figure 10.** (a) Seasonal cycle of median recirculation times north of OSNAP East for water parcels initialised in the upper 250 m of the Irminger Current (***IC***, purple) and Irminger Gyre (***IG***, pink) pathways. The total volume transport of upper ***Irm-Irm*** water parcels flowing southward in the EGC within the upper limb (i.e., above $\sigma_{MOC}$) is given by the solid black line ($V_{EGC}$). Note that water parcel volume transports are plotted according to the month during which they cross OSNAP East southward in the EGC. (b) Example water parcel trajectories flowing northward in the upper 250m of the Irminger Current during February (pink) and August (green) 2000. The associated median recirculation times are also coloured accordingly in a). Way point markers show the locations of water parcels remaining within the boundary current at 60- (circles) and 120-days (squares) following initialisation. Trajectories are superimposed on filled contours of the difference between the mean winter (DJF) and summer (JJA) northward component of the wind stress fields calculated between 1976-2015. The time-mean wind stress field (1976-2015) is shown by the overlying vectors included at the centre of every $4^{th}$ model grid cell. (c) Distributions of upper ***IC*** recirculation times determined from all of the water parcels flowing northward across OSNAP East during February (pink) and August (green) between 1976-2008 (i.e., 33 initialisations for each month).



## 7 Discussion and conclusions

In this study, we investigate the nature of seasonal overturning variability within the eastern North Atlantic SPG using La-
grangian water parcel trajectories evaluated within an eddy-permitting ocean sea-ice hindcast simulation. We employ the
recently introduced Lagrangian Overturning Function (LOF) in density-space, complementing the traditional Eulerian over-
turning streamfunction, to diagnose the net diapycnal transformation integrated along water parcel trajectories traced from the
northward inflows across a model-defined OSNAP East section. By decomposing the LOF according to the individual circula-
tion pathways of the eastern SPG, we determine the principal routes by which the seasonal signal of water mass transformation
north of OSNAP East can imprint onto the strength of overturning recorded at the section. Furthermore, our analysis highlights
the crucial role of water parcel recirculation times within the eastern SPG in driving both Lagrangian and Eulerian overturning
variability on seasonal timescales.

From a Lagrangian perspective, we show that the formation of upper NADW along the cyclonic pathways of the eastern
SPG is strongly dependent upon when water parcels flow northwards across OSNAP East. The lightest water parcels, flowing
northward during autumn, undergo the greatest positive diapycnal transformation due to intense wintertime surface buoyancy
loss along-stream and constitute the largest seasonal volume flux into the lower limb. Meanwhile, the weakest volume flux
into the lower limb is associated with the densest water parcels arriving at OSNAP East in spring and results from negative
diapycnal transformation owing to summertime surface buoyancy gain along-stream. The pronounced seasonal cycle of La-
grangian overturning within the eastern SPG is therefore predicated upon water parcels preserving the seasonal signature of
surface-forced water mass transformation north of OSNAP East. The magnitude of seasonal variability (SD = ±2.2 Sv) re-
mains small compared with the time-mean Lagrangian overturning (8.9 Sv), however, since only water parcels which spend
less than 8.5 months north of OSNAP East are able to avoid irreversible diapycnal transformation into the lower limb. The
majority of northward transport in the upper limb of the LMOC, sourced from the central and southern NAC branches, exceeds
this threshold timescale, crossing the Reykjanes Ridge directly north of OSNAP East in order to reach the EGC within 1-3
years. Pathways crossing the Reykjanes Ridge therefore dominate the mean strength of the Lagrangian overturning since water
parcels are almost guaranteed to form upper NADW along-stream because they experience at least one winter of intense surface
buoyancy loss north of OSNAP East. In contrast, seasonal Lagrangian overturning variability originates from the northward
inflows into the Irminger and Central Iceland Basins, where the shorter path around the cyclonic SPG (≤8.5 months) enables
some water parcels to undergo summertime buoyancy gain and avoid re-densification during the ensuing autumn-winter. The
location of the OSNAP East array therefore has a significant influence on the seasonality of the measured Lagrangian overturn-
ing, by determining the advective timescales on which water parcels recirculate north of the section. For example, deploying
the OSNAP East array further south would likely reduce the amplitude of seasonal Lagrangian overturning variability, given
that a greater proportion of recirculating water parcels would experience at least one winter north of the section and hence be
transferred into the lower limb.

Given the asymmetry between stronger wintertime surface buoyancy loss and weaker summertime buoyancy gain over the
Iceland and Irminger Basins (de Boisséson et al., 2010; Brambilla et al., 2008; Xu et al., 2018b), it is also interesting to frame





seasonal Lagrangian overturning variability somewhat analogously to "Stommel's Demon" (Stommel, 1979; Williams et al., 1995). That is to say, water parcels advected northwards across OSNAP East in the upper limb of the LMOC are participating in a recirculation race against time to avoid wintertime diapycnal transformation into the lower limb. The majority of water parcels are unsuccessful and hence contribute to the mean strength of overturning within the eastern SPG. However, a small

collection of water parcels at a specific time of year recirculate sufficiently quickly to imprint onto the seasonality of Lagrangian overturning downstream. The recent study of MacGilchrist et al. (2021) extended the concept of a seasonal "Demon" by demonstrating that the subduction of water parcels into the interior of the SPNA is additionally modulated by interannual variations in atmospheric forcing, such as the North Atlantic Oscillation (NAO; Hurrell 1995). While it is beyond the scope of this study, it would be interesting to establish whether the seasonal cycle of Lagrangian overturning exhibits a similar sensitivity

to interannual variations in surface buoyancy forcing over the Iceland and Irminger Basins. It may alternatively be the case that enhanced surface heat loss, such as that associated with the strong positive phase of the NAO during the early-1990s (Visbeck et al., 2003; Bersch et al., 2007), would be of secondary importance to changes in water parcel recirculation times owing to the intensification and contraction of the SPG circulation (Curry and McCartney, 2001; Flatau et al., 2003; Pollard et al., 2004).

    Through a detailed investigation of the circulation pathways responsible for the seasonal cycle of Lagrangian overturning

at OSNAP East, we show that 75% of all overturning seasonality originates from the upper 250 m of the Irminger Sea. The remaining seasonality is sourced from the interior recirculation in the upper 250 m of Central Iceland Basin. We identify two dominant modes of seasonal overturning variability according to whether a water parcel circulates along the longer path encircling the boundary of the Irminger Sea ($\bar{\tau} \approx 6$ months) or follows a faster route ($\bar{\tau} \approx 4$ months) within the interior of the Irminger and Central Iceland Basins. Although surface-forced diathermal transformation dominates the along-stream densifi-

cation of both modes, diahaline transformations still play an important role in shaping the character of seasonal overturning variability. Water parcels rapidly recirculating within the Irminger Gyre become lighter (denser) when flowing northward across OSNAP East between February-July (August-Jan) as a result of surface heat gain (loss), which is weakly compensated by year-round salinification owing to lateral mixing with warmer and saltier SPMWs advected along the boundary current. Conversely, water parcels advected along the longer the Irminger Current pathway only become lighter when flowing north-

ward across OSNAP East in May and June. This narrow window of negative diapycnal transformation is principally explained by the enhanced freshening of water parcels downstream in the EGC, consistent with the greater mixing observed between boundary current and Arctic-origin waters along the shelfbreak during winter (Le Bras et al., 2018). Our proposition that the contrasting diahaline changes along the boundary current (freshening) and within the basin interior (salinification) result from mixing is supported by Xu et al. (2018a), who found that, on average, the mixing-induced diapycnal transformation is

negative (lightening) on the inshore side of the boundary current and positive (densification) between the boundary current and the interior. Interestingly, for water parcels flowing northward in the upper 250 m of the Irminger Current in winter, we find negligible diahaline transformation along-stream, indicating that densification is entirely due to surface heat loss along the boundary current. This aligns with previous studies, which have documented the seasonal evolution of boundary currents properties in terms of wintertime diapycnal transformation due to air-sea interaction and summertime along-isopycnal mixing

owing to boundary-interior exchange within the SPNA (Cuny et al., 2002; Huang et al., 2021). While we have only sought to





infer the contributions of air-sea interaction and mixing to the net densification along water parcel trajectories in this study, it would prove valuable for a future study to formally diagnose their respective contributions to the diathermal and diahaline transformations governing the densification of water masses within the SPG.

Our Lagrangian analysis also demonstrates how the longest circulation pathways within the eastern SPG maintain the consistent water mass properties of the lower limb flowing southward across OSNAP East. By examining seasonal water mass transformations along the dominant **Ic-RR-Irm** overturning pathway, we show that, provided water parcels spend at least one winter north of OSNAP East, they will undergo sufficient surface-forced diathermal transformation to form upper ISIW, irrespective of their properties on inflow. We therefore propose that wintertime surface buoyancy loss over the Iceland and Irminger Basins acts as a crucial damping mechanism for seasonal thermohaline variability imported from the NAC upstream. Meanwhile, the stable year-round composition of lower limb waters flowing southward in the EGC results from mixing with Arctic-origin waters along the western boundary of the Irminger Basin. Interestingly, the recent study of Fu et al. (2020) shows that the insensitivity of subpolar overturning to large-scale thermohaline variability (e.g. Holliday et al. 2008; Lozier et al. 2008; Thierry et al. 2008; Holliday et al. 2020; Desbruyères et al. 2021) may also occur on decadal timescales in the eastern SPG. This implies that, given a sufficiently long advective timescale within the eastern SPG, the combination of surface buoyancy loss and interior mixing north of OSNAP East can act as a sink of upper-ocean thermohaline variability and therefore maintain the stability of the MOC.

From an Eulerian perspective, we show that the net transport in the upper limb of the MOC exhibits a pronounced seasonal cycle of 4.1 Sv at OSNAP East, consistent with estimates made using both observations and ocean reanalyses within the eastern SPG (Mercier et al., 2015; Wang et al., 2021). Moreover, seasonality in the strength of Eulerian overturning is closely related to the density structure along the Irminger Sea western boundary. The weakest Eulerian overturning occurs in October when the outflowing EGC is lightest, yielding a large southward transport (-4.9 Sv) in the upper limb of the MOC. We find the largest Eulerian overturning in April when the density structure of the EGC, closely reflecting that of the basin interior, results in the weakest southward transport (-2.1 Sv) within the density classes of the upper limb and the strongest southward transport in the lower limb. This agrees closely with the results of Holte and Straneo (2017), who used Argo profiling floats to show that the MOC seasonal cycle in the Labrador Sea peaks in spring in conjunction with the outflow of newly formed LSW in the Labrador Current. However, in contrast with previous studies which attribute the seasonality of overturning to the seasonal export of western boundary density anomalies alone (Brandt et al., 2007; Holte and Straneo, 2017; Wang et al., 2021), we also highlight the important role of seasonal water parcel recirculation times in the upper Irminger Sea. In addition to the deepening of $\sigma_{MOC}$ along the western boundary of the Irminger Sea, the enhanced upper limb EGC transport responsible for the MOC minimum in October is owed to a convergence of **Irm-Irm** water parcels which flowed northward across OSNAP East in the upper 250 m of the Irminger Basin through spring. This convergence is the result of decreasing recirculation times of upper **Irm-Irm** water parcels north of OSNAP East, which we propose is consistent with a spin-up of the SPG in response to the intensification of basin-scale wind stress forcing during the ensuing autumn-winter (Daniault et al., 2011; Le Bras et al., 2018; Pacini et al., 2020).



This study demonstrates that the advective timescales over which water masses circulate within the eastern SPG plays a critical role in shaping the simulated Eulerian and Lagrangian seasonal overturning variability at OSNAP East. However, what remains unclear is how our principal findings might change in an ocean sea-ice hindcast simulation performed at eddy rich rather than eddy-permitting resolution (e.g., Böning et al. 2016; Marzocchi et al. 2015; Biastoch et al. 2021; Hirschi et al. 2020). On the one hand, we might anticipate that advective timescales will increase with horizontal resolution since improved representation of submesoscale and mesoscale dynamics produces more dispersive water parcel trajectories (Gary et al., 2011). However, the study of Blanke et al. (2012) shows that the inclusion of small-scale structures in high resolution velocity fields reduces water parcel transit times despite increasing the complexity of trajectories. This is further compounded by the stronger SPG circulation simulated in eddy-rich models compared with those configured at eddy-permitting resolution (Jackson et al., 2020; Hirschi et al., 2020), which favours shorter advective timescales and a larger seasonal overturning signal in the eastern SPNA. Establishing whether the mechanisms governing overturning seasonality are consistent across numerical models should therefore be considered an ongoing priority, especially given that the accurate assessment of long-term trends in the strength of the subpolar MOC is predicated on adequately resolving the variability simulated on seasonal timescales.

*Code and data availability.*  The Lagrangian trajectories used in the analysis can be obtained from https://doi.org/10.5281/zenodo.6573900. The Lagrangian trajectory code TRACMASS, developed by Aldama-Campino et al. (2020), is available from https://doi.org/10.5281/zenodo.4337926. Full details of the NEMO ocean model configuration, including access to forcing files, is available on GitHub (https://github.com/meom-configurations/ORCA025.L75-GJM189.git), and has been released with an associated DOI (https://doi.org/10.5281/zenodo.4626012.)

*Author contributions.*  OJT, HLJ and CW defined the overall research problem and methodology. OJT conducted the Lagrangian experiments, performed the analyses and wrote the manuscript. All authors (OJT, HLJ, CW, DGE) discussed and refined the text and contributed to the interpretation of results.

*Competing interests.*  The authors declare that the research was conducted in the absence of any conflicts of interest.

*Acknowledgements.*  OJT is grateful for the financial support of the UK Natural Environment Research Council (NE/S007474/1). HLJ was supported by the NERC-NSF grant SNAP-DRAGON (NE/T013494/1). CW was jointly supported by the NERC LTS-S CLASS (Climate–Linked Atlantic Sector Science) grant (NE/R015953/1) and the NERC LTS-M CANARI (Climate change in the Arctic-North Atlantic Region and Impacts on the UK) grant (NE/W004984/1). DGE was supported by the NERC LTS-S CLASS grant (NE/R015953/1). We would like to thank the European Drakkar project, who carried out the hindcast simulation, and to J. M. Molines and C. Talandier, who kindly provided the data. We additionally thank Laura Jackson, who kindly provided the code to extract the coordinates of OSNAP East array from the ORCA025 model grid.





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
