# Peer review of "Seasonal overturning variability in the eastern North Atlantic subpolar gyre: A Lagrangian perspective"

_EGUsphere, 2022_

## Author Comment (AC1)

**Seasonal overturning variability in the eastern North Atlantic subpolar gyre: A Lagrangian Perspective**

O.J. Tooth, H.L. Johnson, C. Wilson, D.G. Evans

**Responses to Reviewer 1 Comments**

We would like to thank each of the reviewers for dedicating their time to reading the manuscript and providing constructive feedback. We have acted upon all of the comments and suggestions proposed by reviewers, which we believe has led to a significant improvement in the structure and clarity of the original manuscript.

Our response to Reviewer 1 is structured as follows: **Section A** addresses the major comments of Reviewer 1 concerning the methodology of our study and the structure of our manuscript. **Section B** addresses the minor comments of Reviewer 1 on the contents of the manuscript. In both sections, our responses are included in **red** and the original Reviewer comments are included in **blue**.

**Summary**

Using a NEMO-based forced global numerical hindcast simulation, the authors characterize the seasonal variability of the overturning circulation across the OSNAP-east section in the Subpolar North Atlantic. Combining Eulerian and and Lagrangian approaches, they found that such seasonality is critically controlled by the transit time of water parcels north of the section, with a 8.5 month threshold beyond which diapycnal transformation is irreversible (i.e. contribution to the mean overturning, not the seasonality). They further describe the pathways and mechanisms underlying the seasonal signal, and show the key role for wind- driven changes in recirculation time of upper water masses in Irminger Sea. Distinguishing the respective pathways and timescales that characterize the seasonal and mean overturning is a key asset of this work.

It was overall a pleasure to read this manuscript. It is well and precisely written, with a rigorous and very comprehensive analysis of well-posed scientific questions. I think it comprises significant findings and will represent, alongside its already-published companion paper on the mean overturning state, a timely contribution to the field. Therefore, I have only a few general and minor comments, which I list below.

**Section A**

The paper is long, quite dense and detailed, which could give some readers a hard time to eventually extract the key take-home messages. I would suggest shortening the text where possible and only keep the key findings in the main text, and maybe put the complementary diagnostics in supplementary materials. This is more of an advice than a request.

We agree with the reviewer that, on reflection, our original manuscript often favoured detail over the clarity of our central messages and have therefore significantly revised the manuscript to reflect this. In addition to reordering the text to ensure consistency

between the structures of the manuscript sections, we have rewritten the Introduction to present a more concise motivation for our key research questions. We have additionally decided to shorten our original Section 5.2 entitled *Transformation along seasonal overturning pathways* by combining our original Figures 6a-b, 6e-f & 7c-d following the suggestion of Reviewer 3 and moving the original panels (Figures 6c-d) detailing the seasonal density changes along each overturning pathway to the Appendix (see Figure A1).

It should be made clearer (in the abstract notably) that the results apply to the OSNAP-east section only. In some instance, the findings are presented as relevant for the entire eastern SPNA (e.g. line 13-16). Although the sensitivity of the results to the specific section location is mentioned at line 579-584, I think this should be better emphasized throughout the paper. In fact, I wonder if replacing "in the eastern North Atlantic subpolar Gyre" by "across the OSNAP-east section" in the title would be indeed more correct.

Following the reviewer's suggestion, we have significantly revised the abstract and main text to make it clear to readers throughout our study that our analysis is conducted at a model-defined OSNAP East section. We agree with the reviewer that the location of OSNAP East plays an integral role in determining the seasonality of overturning recorded at the section and have modified our discussion to better emphasise this point in the context of the Eulerian observations made along the OSNAP array [Lines 564-568]. We additionally acknowledge that the relationship uncovered between seasonal wind stress forcing and overturning seasonality at OSNAP East may not be applicable elsewhere within the SPG and highlight the important role of basin geometry in modulating this relationship [Lines 319-325]. We prefer to keep the title as is to ensure it remains both appealing and accessible to readers who may not be familiar with the OSNAP array.

Although the authors start to elaborate on the possible impact of using higher-resolution runs at the very end of the manuscript, I believe more could be said on their use of a single numerical model to infer general conclusions about the overturning. I think the authors should better acknowledge in their manuscript that their results might be model-dependent, and specifically point out the most sensitive diagnostic/results accordingly.

We have addressed this by restructuring our discussion of the limitations associated with our study to identify two specific results which we anticipate will be most sensitive to changes in model resolution (horizontal and vertical) and the choices of physical parameterisations. As highlighted in our original discussion, we would expect shorter water parcel recirculation times at eddy-rich / eddy-resolving resolution to favour a larger seasonal Lagrangian overturning signal at OSNAP East. Preliminary results from an ongoing model comparison study, investigating the impact of increasing horizontal model resolution on the strength and variability of Lagrangian overturning across the OSNAP sections, support this hypothesis (the amplitude of the LMOC seasonal cycle increases by ~2 Sv at OSNAP East in a simulation at 1/12 degree resolution). Secondly, we acknowledge that the mixing-induced diapycnal transformations sampled along water parcel trajectories are likely to be strongly dependent upon the representation of (sub)mesoscale mixing and therefore the chosen parameterisation of lateral and vertical eddy mixing in the ORCA025-GJM189 simulation [Lines 626-647]. We also acknowledge the potential role of well-established model biases, such as the excessive entrainment of ambient Atlantic water by the Nordic Seas overflows, in shaping our finding that the composition of the lower limb exhibits remarkable stability at OSNAP East in this simulation [Lines 533-536].

On the same topic, references to previous validation of that particular simulation in the SPNA should be added in Section 2.1 (how well does ORCA025-GJM189 represents the basic features of the subpolar North Atlantic?)

We have added a further paragraph to Section 2.1 to address the fidelity of the subpolar North Atlantic circulation and hydrography simulated by ORCA025-GJM189 as suggested by the reviewer. Since the previous studies of MacGilchrist et al. (2020), Asbjornsen et al. (2021) and Tooth et al. (2023) have all presented detailed validations of the ORCA025-GJM189 simulation within the domain of interest in this study, we choose to summarise their existing conclusions rather than reproduce this analysis [Lines 104-117].

I was left wondering whether the results could be sensitive to the chosen parametrization of turbulent convective mixing within the mixed layer (random perturbation of vertical velocities)? Could the authors comment on this in their Methods section?

We are grateful for this excellent question by the reviewer and have included here a supplementary Figure S1 below to address this directly. As noted on Lines 191-193, the strength and variability of the Lagrangian overturning at OSNAP East is not found to be sensitive to our parameterisation of vertical convective mixing along water parcel trajectories in the surface mixed layer. We determined this by comparing our original monthly time series and seasonal cycle of Lagrangian overturning for the shorter period between 1996-2008 with the equivalent diagnostics calculated from water parcel trajectories evaluated without vertical convective mixing in the surface mixed layer.

[Figure]

**Figure S1.** Lagrangian overturning variability of the eSPG at the model-defined OSNAP East section excluding vertical convective mixing in the surface mixed layer. (a) Monthly maximum of the Lagrangian Overturning Function in density-space with (black line) and without (green line) vertical convective mixing in the surface mixed layer between 1996-2008. (b) Mean seasonal cycles of the maximum Lagrangian overturning within the eSPG computed from monthly composites with (grey shading) and without (green shading) vertical convective mixing in the surface mixed layer. Shading represents +/-1 standard error of the monthly estimates.

**Section B**

65-69: I am not sure to follow the point made here. Comparing the seasonal AMOC amplitude (4 Sv) with that of its potential surface forcing (20 Sv) makes sense (although the volume term hampers this comparison, as stated), but comparing it to the mean strength (16 Sv) is less clear to me.

On revising the Introduction of our manuscript, the sentence highlighted above was removed due to its lack of clarity.

152: Integrating from the sea surface (instead of from the bottom) implies that the MOC strength includes the net transport through the section because one can assume that the northward transport into the Arctic takes place in the shallowest layers. Some studies indeed use bottom-up integration to strictly capture the overturning (in fact the authors here remove this net throughflow to provide the mean at line 190). Therefore, I wonder whether the Eulerian seasonal signal does include a contribution from the seasonal variability of the net throughflow? Was it also removed from the total signal?

We would like to thank the reviewer for posing this intriguing question regarding the decision to exclude or include the net throughflow across the OSNAP East section in our Eulerian overturning analysis. In reflection of its importance, we have added a paragraph in Section 2.3 *(Definitions of the overturning in density-space)* to outline our chosen approach and explain its implications for our central findings [Lines 165-171]. The reviewer is correct that, by electing to integrate the overturning streamfunction from the lightest to the densest isopycnal surface, our definition of the MOC includes the net throughflow across the OSNAP East section. This decision was primarily motivated by our desire to investigate the volume transport structure of the MOC upper limb in Figure 3 (formerly Figure 8) and because, to compensate for the net throughflow, we would have to assume a spatial structure for its distribution along the OSNAP East section which we prefer to avoid. To assess the impact of including a time-varying net throughflow in our Eulerian overturning calculation, supplementary Figure S2 below presents the monthly time series of the maximum Eulerian overturning excluding the net throughflow across the section (i.e., MOC – $V_{net}$) and the seasonal cycles of the MOC both including and excluding the net throughflow contribution. Importantly, we do not find a seasonal signal in the net throughflow across OSNAP East in our hindcast simulation and thus its inclusion in the Eulerian overturning diagnostic does not change the physical mechanisms identified as underpinning seasonal Eulerian overturning variability in our study [Lines 170-171].

[Figure]

**Figure S2.** Eulerian overturning variability at the model-defined OSNAP East section excluding the net throughflow to the Arctic. (a) Monthly maximum of the Eulerian overturning in density-space (first available day of each month, grey line) and 1-year annual running mean (black bold line) overlaid for 1976-2008. The net throughflow across the OSNAP East section is removed from the maximum of the Eulerian overturning streamfunction each month. (b) Mean seasonal cycles of the maximum of the Eulerian overturning computed from monthly composites including (orange shading) and excluding (pink shading) the net throughflow across the OSNAP East section. Shading represents +/-1 standard error of the monthly estimates.

249-250: What explains the 1 Sv difference between the peak-to-peak amplitude of the seasonal MOC (4.1 Sv) and LMOC (5.1 Sv)? This should be explained.

We note that, by definition, the Eulerian (MOC) and Lagrangian overturning (LMOC) represent fundamentally different measures of subpolar overturning variability and we would therefore not necessarily expect their seasonal variability at OSNAP East to be of similar magnitude. The Eulerian overturning streamfunction zonally integrates the volume transports normal to the OSNAP East array in density-space at a given point in

time (the start of each month in our study), whereas the Lagrangian overturning function measures the total diapycnal transformation a water parcel will go on to experience during its recirculation north of the section. As such, the seasonality in the Eulerian overturning can reflect both discontinuities between the instantaneous northward and southward transport components across OSNAP East (e.g., those resulting from seasonal wind-driven circulation changes as shown in Section 3.2) and seasonal changes in the density structure of the upper ocean along the section. In contrast, seasonal Lagrangian overturning variability reflects changes in the along-stream transformation of water masses into the lower limb of the overturning circulation as a function of when they flow northwards across the OSNAP East section. Thus, the amplitude of the LMOC seasonal cycle can be interpreted as follows: an additional 5.1 Sv of the water arriving at OSNAP East in November is transferred into the lower limb compared with the water flowing northward across the section in May. The amplitude of the Eulerian overturning seasonal cycle instead tells us that an additional 4.1 Sv of water flows northward across OSNAP East in the upper limb of the MOC in April compared with October.

348: Vage et al (2011) is an observational analysis, so it is not obvious whether their definition of boundary-interior limit (500 km) applies in the model too. Are the simulated IG and IC characteristic in line with observed ones ?

As discussed in our previous paper (Section 3 in Tooth et al. 2023), the northward inflows of the Irminger Gyre and Irminger Current across OSNAP East are most appropriately distinguished by their sharp density contrast in the ORCA025-GJM189 simulation. The time-mean velocity field (1976-2015) included in the supplementary Figure S3 below shows that, consistent with the observation of Våge et al. (2011), the lighter, surface intensified Irminger Current inflow is confined to the the western flank of the Reykjanes Ridge (500 km < x ≤ 750 km), whereas the inflows to the Irminger Gyre occupy the basin interior (x ≤ 500 km). We have also included a statement on Line 283 that our decision to use the Våge et al. definition of the Irminger Current and Irminger Gyre transports is supported by our simulated time-mean current structure.

[Figure]

**Figure S3.** Time-mean Eulerian velocity field (1978-2015) simulated across the OSNAP East section in the ORCA025-GJM189 ocean sea-ice hindcast. Overlaid black lines represent the time-mean isopycnals of maximum Eulerian overturning (27.52 kg m$^{-3}$)and the observed upper limit of overflow waters in the subpolar North Atlantic (27.80 kg m$^{-3}$).

---

## Author Comment (AC2)

**Seasonal overturning variability in the eastern North Atlantic subpolar gyre: A Lagrangian Perspective**

O.J. Tooth, H.L. Johnson, C. Wilson, D.G. Evans

**Responses to Reviewer 2 Comments**

We would like to thank each of the reviewers for dedicating their time to reading the manuscript and providing constructive feedback. We have acted upon all of the comments and suggestions proposed by reviewers, which we believe has led to a significant improvement in the structure and clarity of the original manuscript.

Our responses are included in **red** and the original Reviewer comments are included in **blue**.

**Summary**

The manuscript describes an in-depth analysis of Lagrangian-derived overturning in the eastern subpolar gyre, with an emphasis on the seasonal cycle. The analysis is conducted wholly in the ORCA025 model run that extends from 1958-2015. The authors compare the Eulerian AMOC in the model to a Lagrangian-derived AMOC by seeding Lagrangian particles in the northward currents across OSNAP East and calculating their net water mass transformation by the time the water particles recirculate southwards across OSNAP East. The authors find that the seasonal cycle across OSNAP East is primarily driven by fast-moving particles that recirculate in the region within 8.5 months.

I found the manuscript fascinating and the figures beautiful. The text is quite long, but each of the sections provided interesting information. I commend the authors for the amount of work and diligence it must have taken to prepare a manuscript with this much material. For this reason, I was torn about how to review this paper: on one hand, the paper is extremely polished, while on the other hand, I found a few major concerns about the manuscript (described below). I have decided to recommend the paper be reconsidered pending major revisions, mostly because my concerns underlie the basis of the paper, and they give me serious pause when trying to learn what to take away from the paper. Without sufficiently addressing these concerns, the manuscript lacks a central message despite the fascinating results along its circuitous journey (which is analogous to the journey these Lagragian particles take around the Iceland Basin...).

**Major Comments**

1. It is unclear to me what the goal of the paper is and/or what signal the authors are trying to explain. Is it the observed seasonality at OSNAP East? Or possibly the model's version of seasonality at OSNAP East? If the authors are trying to explain the observed seasonality at OSNAP East, it has not yet been identified in the published literature, so it seems strange to try to explain it without the signal being identified. Can the authors diagnose the OSNAP East seasonal cycle from the publicly-available data and use that as a motivation for the current study? And if it is the model's seasonality, the authors should explain the importance of a single model's representation of the seasonal cycle, especially whether it resembles the observations.

We have completely rewritten the Introduction and made significant edits elsewhere to make it clear that the goal of the paper is a thorough understanding of what determines seasonal variability in overturning at the model's OSNAP East section. This should help us understand what we see in the observations once that time-series is published. See the response to point 2 below for further details on the changes made.

2. Part of my concern in #1 arises from ambiguity in the introduction – individually the sentences are factually correct and well-written, but I often didn't understand how one sentence led to the next. This is true throughout the section, though I will highlight the sentence starting "It therefore remains an open question..." (line 80) because it is critical to motivating this paper. The previous sentences were discussing seasonal versus interannual variability, then in this sentence the authors shift to comparing the mean AMOC to its seasonal cycle. I generally agree with the sentences individually up to this point, I just don't know how it leads to the authors' question that they seek to address in the paper. I also didn't understand the importance of this question: why does it matter whether the particles that determine the mean state are the same as the ones that determine the seasonality? To address this issue throughout the introduction (not just the example I provided), I recommend the authors highlight a single (or set) of questions that they aim to address in the manuscript, and provide motivation for why those questions are important in the introduction. Otherwise, the text seems to ramble through a lot interesting topics, but lacks clear, identifiable results. I also believe that much of the text could be condensed and made more readable if the goals of the study were outlined early.

On reflection, we can see why the reviewer found significant ambiguity in the aims of our original manuscript and have therefore made extensive revisions to the entire text. While the reviewer is correct to point out that OSNAP was unable to identify a statistically significant seasonal cycle in the Eulerian overturning measured at OSNAP East between 2014-2018, Fu et al. (*submitted*) document a statistically robust observed overturning seasonality determined from the longer 6-year MOC time-series recorded at the section between 2014-2020. With that said, previous observations, such as Mercier et al. (2015) at the OVIDE section, and observationally constrained reanalyses (Wang et al. 2020) have already reported pronounced seasonal overturning variability within the eastern subpolar North Atlantic. Given that the amplitude of such overturning seasonality amounts to a comparatively small fraction of the seasonal buoyancy-driven transformation in this region, we, therefore, seek to understand the extent to which seasonal dense water formation drives seasonality in the overturning evaluated at OSNAP East.

Previous studies, adopting the traditional Eulerian overturning framework, have attributed seasonal overturning variability entirely to seasonal changes in the density structure of the Irminger Sea western boundary current. However, we are able to demonstrate that this explanation fails to recognise how wind-driven seasonal changes in the volume transport of the boundary current might also contribute to the seasonal signal of Eulerian overturning at OSNAP East. Moreover, we should also ask whether the traditional Eulerian framework is best placed to understand the seasonality of overturning, given that it is predicated upon the efficient export of newly ventilated water masses within several months of their formation upstream. We instead argue that adopting a Lagrangian perspective which quantifies the formation of dense water masses along the circulation pathways of the eastern SPG (within the Iceland and Irminger basins) is more appropriate since this explicitly accounts for the wide distribution of recirculation times north of OSNAP East. We now explicitly state the purpose of our study is "to identify the circulation pathways responsible for seasonal overturning variability at OSNAP East and characterise their advective timescales and along-stream transformations within the eSPG" on Lines [66-68].

We have also modified the structure of our manuscript to ensure that the Eulerian and Lagrangian analyses are presented separately to avoid ambiguity in the interpretation of our central results.

3. The authors use ORCA025 exclusively and do not motivate why this would be a good, or even sufficient model to use for this analysis. There are certainly higher resolution models run for similar time periods readily available, so I would hope that there is a reason to use this model over the others (e.g. HYCOM, VIKING, other NEMO-based model runs, ECCO, etc.).

The reviewer was right to highlight the lack of motivation for our use of the ORCA025-GJM189 hindcast in our original manuscript. To address this, we have added two new paragraphs on Lines [97-117] in the Methods section to communicate our reasoning, which can be summarised as follows:

1. This specific ORCA025-GJM189 simulation has been used in previous Lagrangian analyses investigating the circulation pathways of the subpolar North Atlantic Ocean.

2. While there are naturally biases in the circulation and hydrography at eddy-permitting resolution (predominantly at depth where there is a poor representation of the Nordic Seas overflows and hence an absence of deep stratification), the horizontal resolution of the ORCA025-GJM189 model is approximately four times finer than the typical resolution used in the ocean component of CMIP6 climate models. Furthermore, the ORCA025 configuration is more typical of the climate model involved in the previous HighResMIP and upcoming CMIP7 experiments (e.g. the HadGEM3-GC31-MM model included in HighResMIP uses the eORCA025 configuration of NEMO as its ocean component). We argue that it is therefore critical to document the physical mechanisms governing the seasonality of subpolar overturning at this resolution, given that the accurate assessment of long-term trends in the strength of the MOC is predicated on adequately resolving higher frequency variations, including those occurring on seasonal timescales.

3. The ORCA025 configuration has been extensively validated in the subpolar North Atlantic. We have also included a further paragraph highlighting the results of the validation of the ORCA025-GJM189 hindcast previously undertaken by Tooth et al. (2023), as well as the findings of earlier ORCA025-based process studies in the subpolar North Atlantic. [Lines 108-115]

Though resolution is not the only component of a model that determines its quality, I am concerned that a 1/4° resolution model cannot resolve some of the important processes in the eastern subpolar North Atlantic shown in the literature (e.g. Gary et al., 2018; Houpert et al., 2018; Zhao et al., 2018; Devana et al., 2021), specifically the transformation of water, and any vertical velocities, both of which are highly resolution-dependent, yet important to the AMOC. I am also concerned that a 1/4° cannot properly resolve the three currents that enter the Iceland Basin (Holliday et al., 2020), which are critical to understanding the circulation in the region.

Alongside the addition of a summary of previous validations of the ORCA025 model configuration, we have also added further comparisons between the simulated overturning and gyre circulation and relevant observational estimates made in the eSPG. For example, we compare our simulated strength of the MOC, the isopycnal of maximum Eulerian overturning, the net throughflow across the OSNAP East section, historical trends in the MOC strength between the 1970s-1990s and 1990s-2000s to relevant observations in Section 3.1. [Lines 196-215] We have also modified Section 4.1

(formerly Section 3.2) to highlight the fidelity of our simulated time-mean Lagrangian overturning between OSNAP East and the Greenland-Scotland Ridge by comparing this value to observed estimates of the overturning in this region from volume budget calculations [Lines 342-345].

To better acknowledge the limitations of the ORCA025-GJM189 hindcast simulation, we have extended our discussion to highlight two particularly relevant uncertainties for this study. These are: (i) the impact of horizontal resolution on recirculation times and (ii) the impact of both the chosen physics parameterisations (including sub-grid scale) and model resolution on the mixing-induced diapycnal transformations captured along water parcel trajectories. We strongly agree with the reviewer's comment that model resolution alone does not necessarily lead to a more accurate representation of the circulation and water mass properties and would highlight the work of Chassignet et al. (2020) as a particularly notable example. In particular, we emphasise that while eddy-rich simulations undoubtedly yield significant improvements in the fine-scale structural representation of circulation pathways in the SPNA, this should be weighed against their well-documented biases in subpolar hydrography (salinification of the Labrador and Irminger Sea), which imprint detrimentally onto the diapycnal overturning simulated at eddy-rich resolution. Thus, an accurate depiction of the subpolar overturning circulation requires that both the simulated velocity *and* density (potential temperature and salinity) fields are well represented in comparison with observations. [See Lines 626-647].

We would also like to highlight that preliminary results from our ongoing model intercomparison study, investigating the impact of increasing horizontal model resolution on the strength and variability of the Lagrangian overturning simulated at OSNAP West & East, suggest that the principal conclusions of this study remain robust at eddy-rich resolution.

Finally, we have included the simulated time-mean velocity field across OSNAP East (1979-2015) below to directly address the reviewer's concerns regarding the ORCA025-GJM189 hindcast's ability to reproduce the three North Atlantic Current branches entering the Iceland basin (northern and central branches) and Rockall Trough (southern branch).

[Figure]

**Figure S1.** Time-mean Eulerian velocity field (1978-2015) simulated across the OSNAP East section in the ORCA025-GJM189 ocean sea-ice hindcast. Overlaid black lines represent the time-mean isopycnals of maximum Eulerian overturning (27.52 kg m$^{-3}$)and the observed upper limit of overflow waters in the subpolar North Atlantic (27.80 kg m$^{-3}$).

My final concern about the resolution of the model concerns running Lagrangian trajectories through a coarse-resolution velocity field involves a lot of interpolation between points, with the resultant figures (which are beautifully presented) at much higher resolution than the underlying data, and potentially evoking higher confidence in the results than one might otherwise given 1/4° data. The authors acknowledge this

issue in the last paragraph of the paper, but it needs to be addressed in the data and methods section (if not earlier).

We are grateful to the reviewer for highlighting this concern. As explained in Tooth et al. (2023), where we first document the Lagrangian experiment used in this study [Lines 128-130], TRACMASS solves the trajectory path through each model grid cell analytically by assuming that each component of the simulated 3-dimensional velocity field is a linear function of its corresponding direction [i.e., $u = u(x)$]. In this experiment, we evaluated water parcel trajectories using the regular step-wise stationary scheme, which linearly interpolates the ORCA025-GJM189 velocities between successive 5-day mean velocity fields using a series of intermediate time-steps (see Döös et al, 2017 for further details). Thus, by assuming that the resulting velocity field remains stationary during intermediate steps, an exact solution of the resulting differential equations can be found, representing the streamlines within each grid cell. The only exception is within the surface mixed layer, where we parameterise the effects of vertical turbulent mixing along water parcel trajectories. The trajectories appear very detailed in our Figures because we chose to output the positions along each water parcel trajectory at every model grid cell crossing so that we could analyse the properties of water parcels on their westward crossing of the Reykjanes Ridge in Tooth et al. (2023). To make this clearer to readers, we have now added details of the output frequency of water parcel positions and properties along their trajectories on Lines [146-147].

Again, I want to underscore how impressed I was with the quality of the paper, which reflects quite strongly on the authors.

---

## Author Comment (AC3)

**Seasonal overturning variability in the eastern North Atlantic subpolar gyre: A Lagrangian Perspective**

O.J. Tooth, H.L. Johnson, C. Wilson, D.G. Evans

**Responses to Reviewer 3 Comments**

We would like to thank the reviewers for dedicating their time to reading the manuscript and providing constructive feedback. We have acted upon all of the comments and suggestions proposed by reviewers, which we believe has led to a significant improvement of the original manuscript.

Our response to Reviewer 3 is structured as follows: **Section A** addresses the major comments of the reviewer concerning the Abstract, Introduction and general structure of the manuscript. **Section B** addresses the minor comments of Reviewer 3 on the contents of the manuscript and the accompanying Figures. Our responses are included in **red** and the original Reviewer comments are included in **blue**.

**Summary**

The manuscript by Tooth et al. presents a detailed and thorough model analysis of the seasonal overturning variability happening between the OSNAP array and the Greenland-Scotland Ridge. Combining insights from a Eulerian analysis at the OSNAP array and a Lagrangian framework where the sensitivity of the transformation to inflow characteristics is tested, the authors show that the majority of the seasonality in the overturning is due to water parcels that exhibit a relatively short (< 8.5 months) recirculation time within the eastern North Atlantic Subpolar Gyre.

The analysis is scientifically sound, very well embedded in the existing literature and is valuable to e.g. better interpret OSNAP measurements and improve our understanding of water mass transformation processes. However, I do agree with most of the comments of the other reviewers. The paper is very long, and, due to the smart but tricky to understand Lagrangian method, it can be challenging for the readers to fully grasp the content. Also, the motivation of the study can be more clearly defined in the abstract and introduction. Therefore, I hope to provide some recommendations and suggestions to improve the readability of the manuscript and would advise minor revisions before publication, but with sufficient revision time to restructure the paper.

**Section A**

1. **Abstract**

   More clearly state the motivation / current lack of knowledge in your abstract and how your approach provides new insight. E.g. One of your main findings is that you are only able to explain the minimum MOC in autumn seen in the OSNAP measurements if you use a Lagrangian approach ("This convergence of southward… wind forcing"). This is at the moment not clear in your abstract.

We agree with the reviewer that the original Abstract did not clearly define the gap within the existing subpolar overturning literature that we seek to address in our study. We, therefore, decided to considerably revise the Abstract to highlight one of the central motivations of our study, namely, our inability to attribute seasonal overturning variability to seasonal dense water formation owing to the diversity of recirculation timescales found within the Iceland and Irminger basins. Following the reviewer's suggestion, we have also explicitly stated that the autumn minimum in the MOC seasonal cycle can be explained by adopting a Lagrangian perspective.

The statement "recirculation race against time" is nice to mention in the paper, but might confuse readers in the abstract. And I don't think you need it in the abstract, as it is already clear from the last sentence what your main finding is ("The seasonality of Lagrangian overturning... in the eastern SPG").

Following the reviewer's recommendation, we have removed the reference to our analogy of a water parcel recirculation race against time from the Abstract, and this is now only included in the Discussion and Conclusions section [Lines 583-593].

**2. Introduction**

Try to get to the main research question / motivation for this study within the first two paragraphs. I find some hint for motivation in Ln.78, but I would suggest to get to this much quicker, and state clearly how this is related to the research question and approach of your study.

In general, this introduction can be shortened quite a bit, there are many details that are not needed to understand the motivation of the study, and some can be moved to relevant parts in the manuscript.

[Addressing both of the comments above]
We strongly agree with the reviewer that our original manuscript's Introduction favoured detail on the background of the subpolar overturning circulation rather than concisely motivating the need to bring a complementary Lagrangian perspective to the emerging discussion of seasonal overturning variability within the SPNA. We, therefore, decided to completely rewrite the Introduction and follow the suggested approach of Reviewer 2 to identify a series of key research question(s) that we will go on to address throughout our study.

Ln. 35-39 not necessary in the introduction (can move that to the method section where you explain how you define overturning)

In order to condense the manuscript and emphasise the most important concepts, we decided to remove our original discussion outlining the reasons for favouring a density- versus depth-space overturning definition in the SPNA.

**3. General structure of the paper**

The structure in the abstract differs from the general structure of the paper, and I think it makes a bit more sense to indeed first fully discuss the insights from the Eulerian analysis, before moving on to the Lagrangian one. That would also help to more clearly state what the added benefit is for looking at the relevant mechanisms from a Lagrangian perspective. So e.g. move largest part of section 6 to follow 3.1. Or, have a full section 3 focused on the Eulerian perspective where you have a dedicated section for validation (what now is mainly section 3.1) to also argue why the model you're using is the right choice to address the seasonal variability and related mechanisms. Furthermore, it would make the interpretation of the Lagrangian

results a lot easier when readers have seen the general Eulerian flow structure in this region and the full overturning characteristics from a Eulerian perspective.

We are grateful to the reviewer for this helpful suggestion and have restructured the manuscript and Discussion and Conclusion section therein to match the structure of the Abstract. As such, Section 3 of the manuscript is now dedicated to exploring the seasonal Eulerian overturning variability simulated at OSNAP East and addressing the physical mechanisms responsible, including the important role of seasonal water parcel recirculation times in the upper Irminger Sea. Then, Section 4 introduces our complimentary Lagrangian measure of seasonal overturning variability at OSNAP East before diagnosing the circulation pathways, advective timescales and transformations contributing to the mean strength and seasonality of Lagrangian overturning.

The different seasonal cycle seen when comparing the Eulerian to the Lagrangian framework can be explained a little better. I'm not sure whether readers fully understand this. Maybe as a thought experiment, think what would happen when you would define the Lagrangian overturning "backwards". So, tracing the Southward flow backwards, and define the LMOC overturning in that way. This would again change the seasonal variability observed as you would focus on the seasonality of the outflow, instead of the inflow.

We recognise the reviewer's concerns regarding the comparison between the complementary Eulerian and Lagrangian overturning frameworks used in our study. We believe that much of this confusion will likely have originated from the original structure of the manuscript which did not clearly motivate the need to investigate the overturning seasonality at OSNAP East in our complementary Lagrangian overturning framework in the Introduction, and also failed to separate Eulerian and Lagrangian analyses to avoid ambiguity in their interpretations. On revising the structure of the manuscript and consolidating each of our results sections to better emphasise our most important findings, we believe that a clearer distinction is now made between the Eulerian and Lagrangian frameworks. The reviewer raises an interesting question regarding the decision to trace water parcels forward-in-time from the northward inflows versus backward-in-time from the southward outflows across OSNAP East; however, we would like to highlight two of the reasons underpinning our decision to define the Lagrangian overturning "forwards" in this study:

Firstly, by quantifying the flux of water parcels which are transferred from the upper to the lower limb after their northward crossing of OSNAP East, we ensure consistency with the traditional Eulerian overturning definition used throughout Section 3 (i.e., the net transport of the upper limb of the MOC across OSNAP East, as given by integrating from the lightest to the densest isopycnal surface). This is because our Eulerian and Lagrangian overturning (stream)functions, defined at time $t$, share the same northward volume transport distribution in potential-density space (V_North) and this transport overwhelmingly occupies the upper limb of the overturning circulation at OSNAP East.

Secondly, our decision to define the seasonality of Lagrangian overturning forward-in-time does not influence its physical interpretation as the seasonal signal of dense water formation along water parcel trajectories recirculating north of OSNAP East. If we were to have instead traced the water crossing OSNAP East southwards backward-in-time, we would still have ascribed the seasonal maximum to the stronger flux of water parcel into the lower limb due to intense surface buoyancy loss during the preceding (rather than ensuing) winter months and the seasonal minimum to the weaker volume flux into the lower limb when the fastest recirculating water parcels gain buoyancy along-stream during the preceding (rather than ensuing) summer. By evaluating the Lagrangian overturning "forwards", we are therefore

asking: what fraction of water parcels are transferred into the lower limb during their recirculation, given that they arrived at OSNAP East in the upper limb at the start of month $m$? We would argue that this is more meaningful than asking:what fraction of water parcels are transferred into the lower limb during their recirculation, given that they were exported southward across OSNAP East in the lower limb at the start of month $m$? since we have shown that the potential density of water parcels on their initial northward crossing of OSNAP East is a strong predictor of their future contribution to the Lagrangian overturning at the section.

Check the length of your paragraphs, some of them are extremely long. Try to keep it to one or two main take-aways per paragraph, and keep them in general short (e.g. max length ~12 lines in the current template format).

We recognise that many of the paragraphs contained within our original manuscript were highly congested with results and have therefore revised significant portions of the main text to emphasise the most important findings to be drawn from our analysis. For the most part, paragraphs in the main text are now below the 12-line threshold recommended by the reviewer.

**4. Discussion and conclusions**

Currently the focus is too much on conclusions, and the relevance and importance of the results can be more strongly communicated.
Maybe also put your findings more in the context of the OSNAP observations.

We agree with the reviewer that the Discussion and conclusions of our original manuscript focused heavily on summarising the central results of each of the results sections and did not sufficiently address the importance of our findings to our understanding of subpolar overturning. In addition to restructuring the Discussion to reflect the wider manuscript's structure (seasonal Eulerian overturning variability at OSNAP East now precedes our discussion of seasonal Lagrangian overturning variability), we have now highlighted three important implications of our results in the context of the OSNAP observations.

1. We now emphasise that the proximity of the OSNAP East array to regions of dense water formation (that is, those regions which can transform water masses from the upper to the lower limb of the MOC) exercises an important influence on the seasonality of Eulerian overturning measured at the section. Thus, any future modification of the location of the OSNAP East array may determine the strength of the seasonal signal measured at the section.

2. We have improved our discussion of how interannual modes of variability, such as the North Atlantic Oscillation, might impact our analogy of a water parcel recirculation race against time to avoid irreversible diapycnal transformation into the lower limb. In particular, we hypothesise that the amplitude of seasonal Lagrangian overturning variability at OSNAP East would increase in response to strong positive phases of the NAO, given that corresponding westward retreat of the Subarctic Front favours greater inflow to the Irminger and Central Iceland basins, which dominate the seasonality of Lagrangian overturning in our study.

3. We interpret our Lagrangian analysis in the context of the observational study of Fu et al. (2020), which shows that the strength of the MOC has remained stable over recent decades in spite of large-scale thermohaline variability in the SPNA. We propose that, given a sufficiently long recirculation time within the eSPG, the combination of surface buoyancy loss and interior mixing north of OSNAP East can act as a sink of upper-ocean thermohaline variability advected along water parcel trajectories and thereby maintain a consistent volume flux in the lower limb.

We are currently exploring this final implication in more detail within a hierarchy of ocean hindcast simulations ranging from eddy-parameterised to eddy-rich resolutions as part of a follow-up study which seeks to better understand the stability of the subpolar MOC on decadal timescales.

**Section B**

1 - Why MOC and not AMOC?

While we recognise that the use of the AMOC acronym is more faithful to the Atlantic Meridional Overturning Circulation, our decision to shorten this to MOC throughout the manuscript was primarily motivated by our intention to distinguish this traditionally Eulerian diagnostic from the use of LMOC to refer to our complementary Lagrangian overturning diagnostic. It should also be noted that both Lozier et al. (2019) and Li et al. (2021) use MOC to refer to the Eulerian overturning observed across the OSNAP array.

2 – I find the SPG abbreviation confusing (maybe change to ESPG?), in particular for people that only read the abstract. Even when mentioning the eastern part of the Subpolar Gyre I have a bigger area in mind than the one North of OSNAP and south of GSR. I think the region of interest should be more clearly defined already in the abstract. 7 – Also here, it is not clear for the reader where exactly you are defining this seasonal cycle (minimum AMOC), I do think you should mention the OSNAP array in the abstract.

We have modified the abbreviation of the eastern Subpolar Gyre to eSPG both in the Abstract [Lines 1-2] and throughout the manuscript as suggested by the reviewer. To better orientate readers as to the domain of interest in our study, we have also made reference to the Iceland and Irminger basins on Lines 2-4. The OSNAP array is now referenced throughout the Abstract and our findings are defined in terms of the OSNAP East array rather than the wider eSPG region.

142 – How did you define the Greenland-Scotland Ridge in your model?

The Greenland-Scotland Ridge was defined arbitrarily within our model to connect the shallowest grid cells between East Greenland – Iceland, Iceland – Faroes and the Faroes to Scotland. Although we recognise that our model-defined Greenland-Scotland Ridge section is considerably further south than observational sections, such as the Kögur section north of Denmark Strait, we would like to emphasise that our intention here is simply to determine where we should remove water parcels which travel northwards across the sills.

2 – If I understand the calculation correctly, it should be V_south(sigma, t<tau<tau_max) , to make clear that any parcel that returns within this period of 7 years is added to the LMOC?

We apologise to the reviewer for the confusion caused by our equation for the Lagrangian Overturning Function. Since the calculation represents an integral over all recirculation times, the equation should have included tau (water parcel recirculation time following initialisation) rather than tau_max. We have now corrected this and also added a brief description of the calculation on Lines [177-179] to make this clearer to readers.

220-225 – You could already make a link here why you need a Lagrangian framework to explain why this is the case (now this text might insinuate that already cold and dense waters transported Northward somehow lead to maximum overturning strength).

We are grateful to the reviewer for raising this excellent point. Since our primary motivation for adopting a Lagrangian overturning framework is the need to explicitly account for the diverse range of recirculation times within the eSPG in our revised manuscript, we have decided to make this observation in our discussion of Lagrangian overturning seasonality on Lines [353-354].

243 – Confusing what the transports mentioned in the brackets are.

We have removed the sentence in question which made reference to both Eulerian and Lagrangian model transports. This is both for editorial reasons since we chose to condense Section 4.2 (formerly Section 3) in our revised manuscript, and because we believed that it would be more valuable to contextualise the meaning of the time-mean Lagrangian overturning with Eulerian observations of the overturning within the Iceland and Irminger basins [Lines 342-345].

244 – 243 – I don't understand what is meant here with 'close correspondence', is that somehow visible in one of the figures? Did you calculate a correlation?

We agree with the reviewer that the term 'close correspondence' is ambiguous in the context of this sentence. We have therefore included the correlations between the LMOC seasonal cycle and the seasonal cycles of sigma_MOC and sigma_LMOC and their corresponding p-values (both $p < 0.01$) on Lines [350-353].

267 and elsewhere. In general I think care should be taken when talking about seasons in relation to the LMOC definition as there is a time lag involved in the actual calculation (e.g. when the transformation of the water masses occurs), so it is very difficult to interpret what a minimum in May actually means.

We are grateful to the reviewer for highlighting this concern. We have modified the manuscript to ensure that where we describe the LMOC in a given month it is clear to readers that this corresponds to the month that water parcels flow northwards across the OSNAP section. Throughout the text, we have also made it clear that the strength of the LMOC should be interpreted as the volume flux from the upper to the lower limb integrated along recirculating water parcel trajectories (e.g. Lines [365-369]).

277 – "in contrast", how does the context of this sentence is in contrast with the previous one?

We have revised this sentence and removed "in contrast" as suggested by the reviewer.

302 – The recirculation time itself is probably also seasonally variable? Maybe already address that here?

We have added a sentence in Section 4.2 *Timescales and origins of seasonal Lagrangian overturning* to acknowledge that the recirculation times of water parcels flowing northward across OSNAP East in the upper 250-m of the Irminger and Central Iceland basins exhibit seasonality (as discussed in Section 3.2 on Mechanisms of seasonal Eulerian overturning variability), however, this seasonality does not influence their contribution to the mean seasonal cycle of Lagrangian overturning. [Lines 404-408]

316 – Unclear sentence, which is better explained in the following sentences. Maybe just say "We have identified a threshold recirculation time of 8.5 months". And then continue with explaining what happens to particles < recirculation time, and then > recirculation time.

We have revised the unclear sentence to instead state: "To summarise, we have found a clear distinction between the origins and advective timescales of water parcels responsible for the mean strength and seasonality of Lagrangian overturning at OSNAP East.". The paragraph then continued to outline the nature of this distinction for water parcels contributing to the seasonality and mean strength of Lagrangian overturning, respectively. [Lines 409-415]

330 and elsewhere – I would change the abbreviation of this pathway to Ic-Irm and Ro-Irm, as all pathways are defined by their entry and exit locations and not by crossing RR. Makes it easier to remember for the reader.

We are grateful to the reviewer for this simplification and have renamed both circulation pathways crossing the Reykjanes Ridge as suggested above both in our manuscript text and the accompanying Figures. [Lines 422-425]

335 – "70%" make clear where the reader can find this result and in which figure.

We determine that 70% of the time-mean strength of the Lagrangian overturning at OSNAP East is sourced from the Ic-Irm and Ro-Irm pathways from Figure 8a (now referenced on Line 428. This comes from (4.8 Sv + 1.4 Sv) / 8.9 Sv, where 4.8 Sv is the volume flux into the lower limb along the Ic-Irm pathway (orange) and 1.4 Sv is the equivalent volume flux along the Ro-Irm pathway (red).

 Section 5.2 – This section is extremely long, and due to all the different decompositions very difficult to keep track of what is happening. I would suggest to restructure this, shorten, and maybe split in different sub-sections if needed.

We agree with the reviewer that, on reflection, Section 5.2 on the Transformations responsible for seasonal Lagrangian overturning variability favoured excessive detail in the presentation of our results rather than focusing on the most insightful findings addressing our original research question. We have therefore improved the Section (now 4.4) by implementing many of the suggestions made by the reviewer. These include:

- Consolidating our original Figures 6 & 7 into a single Figure 9 which presents the seasonal cycles of the net diapycnal transformation along the Irminger Gyre, Irminger Current and Ic-Irm pathways together. We elected to remove Figure 6c-d to supplementary Figure A1, alongside the equivalent seasonal cycles of potential density on inflow and outflow crossings of OSNAP East for the IcRo-IcRo and Ic-Irm pathways (including RR crossing as previously shown in Figure 7a).

- Given that Section 4.4 is now followed by the Discussion and conclusions of our study, we have decided to remove the lengthy synthesis of our findings on Lines 467-480 of our original manuscript. Although this was primarily an editorial decision to reduce repetition within our manuscript, we also no longer found it necessary to undertake multiple summaries of our findings after improving the preceding text in Sections 4.3-4.4 to better emphasise our most impactful results.

In total, the changes outlined above have shortened the text in Section 4.4 (formerly 5.2) from 110 lines to 83 lines.

515-519 – This is one of your key findings, the built-up to this result can be made clearer, by already talking about Wang's conclusions in the introduction and stating why these might be insufficient arguments?

We would like to thank the reviewer for this excellent suggestion. On reordering the manuscript to fully discuss the seasonal Eulerian overturning at OSNAP East prior to undertaking a complementary Lagrangian analysis of overturning seasonality, we decided to dedicate the third paragraph [Lines 44-52] of our Introduction to motivating why seasonal density changes alone might be insufficient to account for Eulerian overturning seasonality at OSNAP East. We also emphasise that no study to our knowledge has explored how seasonal variations in the transport and density structure of the Irminger Sea western boundary current might act in concert to modulate the seasonal signal of Eulerian overturning measured across OSNAP East.

**Figures**

Figure 1

I could not find in the text where you reference panel 1b. Also, refer to 1a relatively early in your introduction, to make clear how you define the Eastern SPG (and maybe then use as abbreviation "ESPG" instead of "SPG").

We have used the abbreviation eSPG for the eastern subpolar gyre throughout both the Abstract and manuscript text as suggested by the reviewer. Figure 1a is referenced and the eSPG is defined geographically as the region south of the Greenland-Scotland ridge on Lines [27-29]. Figure 1b is now explicitly referenced in our definition of Lagrangian overturning on Lines [175-177].

Caption: " volume transports across *the model-defined* OSNAP East *array*." (also in caption Fig. 2 and Fig. 3)

We have modified all relevant Figure captions to refer to a model-defined OSNAP East array as suggested by the reviewer.

Dotted line panel b at 1990 does not seem necessary for the storyline.

We have removed the dotted line referring to January 1990 (the year chosen for the example trajectories in Figure 1a) and renamed Figure 1a to Example trajectories initialised across OSNAP East. The chosen seeding time of the example trajectories presented (January 1990) is still referred to in our Figure caption as in our original manuscript.

Figure 4

Panel b, it would be good to add the main flow features already here (not wait until Fig 8d)

On reordering the text, Figure 4 has now become Figure 8 in our revised manuscript. As a consequence, the main flow features across OSNAP East are now first highlighted in Figure 3(d), which decomposes the accumulated upper limb volume transport across the section during April and October (corresponding the extrema of the MOC seasonal cycle at OSNAP East). We have, however, decided to add labels to Figure 8c to identify the major northward inflows across OSNAP East discussed in the main text (Irminger Gyre, Irminger Current and three NAC branches simulated in the ORCA025-GJM189 hindcast).

Panel b, unclear what the thick black line represents.

We thank the reviewer for highlighting this potential source of ambiguity. The solid black line represents the time-mean isopycnal of maximum Lagrangian overturning, sigma_LMOC, which is now labelled in Figure 8b.

Use of white region vs. white line is confusing, also because the white line is missing in your colorbar. I do like the use of the white line to indicate the threshold-time, but then maybe use gray for the masked areas?

We agree with the reviewer that the use of white colouring for both the regions without northward inflow and the spatial limit of the 8.5-month threshold recirculation time along OSNAP East was a potential source of confusion in the original manuscript. We have therefore implemented both of the reviewers' suggestions in Figure 8b; adding a white reference line in the colorbar and masking southward currents across OSNAP East in light grey. The Figure caption has also been modified accordingly.

Figure 5

You start introducing different decomposites based on pathway at the beginning of section 5.1, but then continue to further decompose them (e.g. IC and IG, and the eastern and western part of the IC-Irm pathway). Maybe it would be good to directly make this clear in one figure, so readers can better follow the story and understand the choices made?

We agree with the reviewer that the successive decompositions of circulation pathways north of OSNAP East undertaken in our original analysis did not facilitate a clear and concise discussion of the along-stream transformations responsible for seasonal Lagrangian overturning variability. As suggested by the reviewer, we have consolidated our circulation pathway decomposition to define five pathways, including the Irminger Gyre and Irminger Current, in Figure 8 (formerly Figure 5).

Specify Ic and Ro separately in the figure, now it is unclear how you kept these two pathways separate in your analysis. Maybe, as the decomposition is based on in- and outflow location, add those separation lines in panel b instead with all relevant abbreviations and switch the two panels (b = a). In panel (b) also add 'Reykjanes Ridge'.

We have modified Figure 8 (formerly Figure 5) to address each of the reviewer's suggestions. Figure 8a now defines the Ic and Ro regions alongside the example trajectories for each circulation pathway. We have updated Figure 8b to include three vertical reference lines at 500 km, 750 km and 1475 km, which define the Irminger Gyre

– Irminger Current boundary, Reykjanes Ridge and Ic – Ro boundary along the OSNAP East section. We have also added a label defining the Reykjanes Ridge as suggested.

Use of color, check the 'colorblind' rules, I think the difference between red and orange is very difficult to distinguish (especially in panel b).

We have modified our choice of colours in Figure 8 to ensure a greater contrast between the Ic-Irm and Ro-Irm pathways.

Figure 6 & 7

What do the colored boxes represent? Again, difference between the purples and blue not well visible. Panel c and d can be left out, as this is also seen in e and f. I would merge this figure with Fig. 7, and panel 7b is not needed. Also make clear why you only look at Ic – Irm, and not Ro – Irm.

Following the reviewer's suggestion, we have merged Figures 6-7 in our original manuscript to form a single Figure 9 which presents the mean seasonal cycles of the transport-weighted mean net change in potential density of the upper Irminger Gyre (IG, previously Figure 6e), upper Irminger Current (IC, previously Figure 6f) and upper Ic-Irm pathways (decomposed into the successive transformations taking place in the Iceland and Irminger basins – previously Figures 7c-d). The coloured boxes define each of the aforementioned pathways, which include only water parcels initialised within the upper 250 m of the OSNAP East section. Since we do not discuss the seasonal cycle of diapycnal transformation along the IcRo-IcRo pathway in the manuscript text, we decided to remove the blue boxes previously included in Figures 6a-b (now Figures 9a-b) and hence the distinction between these colours will no longer be problematic.

Figure 8
As mentioned earlier, move this figure to section 3a.

Figure 8 alongside accompanying Figures 9-10 in our original manuscript have now been moved into Section 3.2 – Mechanisms of seasonal Eulerian overturning variability – as suggested by the reviewer above.